# FGF8-mediated gene regulation affects regional identity in human cerebral organoids

Michele Bertacchi[1]*, Gwendoline Maharaux[1], Agnès Loubat[1], Matthieu Jung[2], Michèle Studer[1]*

[1]Univ. Côte d'Azur (UniCA), CNRS, Inserm, Institut de Biologie Valrose (iBV), Nice, France; [2]GenomEast platform, Institut de Génétique et de Biologie Moléculaire et Cellulaire (IGBMC), Illkirch, France

**Abstract** The morphogen FGF8 establishes graded positional cues imparting regional cellular responses *via* modulation of early target genes. The roles of FGF signaling and its effector genes remain poorly characterized in human experimental models mimicking early fetal telencephalic development. We used hiPSC-derived cerebral organoids as an *in vitro* platform to investigate the effect of FGF8 signaling on neural identity and differentiation. We found that FGF8 treatment increases cellular heterogeneity, leading to distinct telencephalic and mesencephalic-like domains that co-develop in multi-regional organoids. Within telencephalic regions, FGF8 affects the antero-posterior and dorsoventral identity of neural progenitors and the balance between GABAergic and glutamatergic neurons, thus impacting spontaneous neuronal network activity. Moreover, FGF8 efficiently modulates key regulators responsible for several human neurodevelopmental disorders. Overall, our results show that FGF8 signaling is directly involved in both regional patterning and cellular diversity in human cerebral organoids and in modulating genes associated with normal and pathological neural development.

*For correspondence:
Michele.BERTACCHI@univ-cotedazur.fr (MB);
Michele.STUDER@univ-cotedazur.fr (MS)

## Editor's evaluation

This study is of interest to neuroscientists interested in brain development, particularly human brain development. Human organoids are excellent models to investigate the relevance of gene and pathways in the context of embryonic development. In this important research paper, the authors present convincing evidence supporting a role of FGF8 in brain development.

## Introduction

The embryonic development of the mammalian brain is orchestrated by a highly coordinated cascade of cellular and molecular events. This cascade engages distinct signaling pathways activated by gradients of diffusing morphogens, including FGFs, BMPs, SHH, and WNTs (*Borello and Pierani, 2010*; *Takahashi and Liu, 2006*; *Rubenstein, 2008*; *Hoch et al., 2009*; *Guillemot and Zimmer, 2011*). These morphogens instruct neural progenitors with spatial and temporal 'coordinates' along the dorsoventral (D/V) and anteroposterior (A/P) brain axes, modulating early expression gradients of key developmental genes in a dose-dependent manner. These initial stages lay the foundation for further critical processes in brain development, such as progenitor proliferation, cell migration, neuronal differentiation, and circuit formation. Disruption of any of these mechanisms can lead to multiple morphological abnormalities, and contribute to a range of neurodevelopmental disorders (NDDs).

**eLife digest** Healthy brain development in the human embryo relies on the precise coordination of numerous molecular signals that guide the formation of distinct brain regions in their correct locations. Molecules that diffuse through embryonic tissues, known as morphogens, serve as spatial and temporal cues that help cells determine their position within the developing brain. These positional signals are crucial for the proper formation of specific brain regions along the embryo's principal axes.

FGF8 is a well-characterized morphogen that influences the anterior-to-posterior regional identity of brain cells in model organisms such as mice. However, studying this process in human embryos poses both technical and ethical challenges, meaning that little is known about the molecular bases of how developing brain cells determine their position along different axes. Understanding these molecular mechanisms is essential for gaining insights into human brain function and the origins of neurodevelopmental disorders.

Bertacchi et al. developed a new 'organ-in-a-dish' system – also known as an organoid – using human induced pluripotent stem cells. The research team combined 2D cell cultures on flat surfaces with 3D tissue culture techniques to create a more reproducible cerebral organoid protocol. This approach enabled the investigation of the role of FGF8 in human brain development within a controlled laboratory setting. Treatment with FGF8 enhanced brain cell diversity, as measured by gene expression analysis through single-cell RNA sequencing. Notably, distinct regions resembling the forebrain (telencephalon) and the midbrain (mesencephalon) emerged in FGF8-treated organoids. Within the telencephalic region, the cell type composition shifted, favoring neurons typically found in the ventral (lower) parts of the human brain. This altered the activity of the neural network, as evidenced by direct electrical signal measurements.

Overall, Bertacchi et al. demonstrated that a single molecular signal, FGF8, can drive the formation of distinct brain regions along multiple axes in human brain organoids. They also identified genes regulated by FGF8 that are associated with neurodevelopmental disorders. One such gene, NR2F1, is well-studied for its involvement in conditions such as intellectual disability, autism and epilepsy. This work provides a biologically accurate cell culture model, offering a valuable tool for advancing research into human brain development and associated neurological diseases.

FGF ligands and their receptors play pivotal roles in neural plate patterning (*Guillemot and Zimmer, 2011*), and disruptions in FGF signaling pathways have been implicated in various brain malformations (*Turner et al., 2016*) and significant cortical development defects (*Sun et al., 2023*; *Shin et al., 2004*; *Hébert et al., 2003*). Notably, although some functional redundancy exists among FGF factors (*Turner et al., 2016*), FGF8 stands out as a primary regulator of regional patterning and brain development. During early development, FGF8 is essential in establishing the midbrain-hindbrain boundary (*Harada et al., 2016*; *Lee et al., 1997*) - a region known as the isthmus, which acts as a brain organizer - where the FGF8 diffusing gradient controls the patterning of posterior brain regions (*Lee et al., 1997*; *Mason et al., 2000*; *Ye et al., 1998*; *Crossley et al., 1996*). During the development of telencephalic vesicles, FGF8 diffuses from the anterior neural ridge (ANR; *Toyoda et al., 2010*; *Hoch et al., 2015*), located at the most anterior pole, creating a gradient that regulates proliferation, apoptosis, regionalization, and identity acquisition along cortical axes (*Toyoda et al., 2010*; *Storm et al., 2006*; *Storm et al., 2003*; *Fukuchi-Shimogori and Grove, 2001*; *Sato et al., 2017*; *Cholfin and Rubenstein, 2007*; *Cholfin and Rubenstein, 2008*; *Garel et al., 2003*). Consistently, FGF8 diffusion is essential for the regulation of key telencephalic genes such as *Foxg1* (*Shimamura and Rubenstein, 1997*), a transcription factor required for the regionalization and growth of telencephalic vesicles. Additionally, FGF8 promotes cortical expansion by activating the ERK signaling pathway, thereby enhancing self-renewal and increasing the pool of cortical radial glia cells (*Sun et al., 2023*). FGFs exert their functions by binding to four highly conserved FGF receptors (FGFRs; *Klimaschewski and Claus, 2021*); among those, FGFR3 plays a specific role in regulating brain size by influencing progenitor proliferation and apoptosis (*Inglis-Broadgate et al., 2005*) and is essential for the correct formation and regionalization of the occipitotemporal cortex (*Thomson et al., 2009*; *Thomson et al., 2007*).

Given their multiple roles during embryonic (*Guillemot and Zimmer, 2011*; *Rubenstein, 2011*) and adult age (*Turner et al., 2016*; *Klimaschewski and Claus, 2021*), it is not surprising that FGF

factors have been implicated in the onset of several NDDs (*Amaral et al., 2008*), and adult psychiatric conditions, such as anxiety, depression, and schizophrenia (*Klimaschewski and Claus, 2021*; *Stevens et al., 2010*). FGF8 exerts its effects by modulating key developmental genes, which, in turn, directly or indirectly control critical neurogenesis processes. Dysregulation of these genes can lead to various pathological conditions. A notable example is the nuclear receptor *Nr2f1*, which is regulated by the FGF8 pathway and plays several roles during mouse telencephalic development (*Bertacchi et al., 2019*; *Tocco et al., 2021*). Mutations in the *NR2F1* gene lead to an emerging NDD, called Bosch-Boonstra-Schaaf optic atrophy syndrome (BBSOAS), a condition characterized by cognitive and visual impairments (*Chen et al., 2016*; *Bosch et al., 2014*; *Rech et al., 2020*; *Bertacchi et al., 2022*). The anterior FGF8 telencephalic gradient negatively regulates *Nr2f1* during early mouse development (*Sansom et al., 2005*; *Assimacopoulos et al., 2012*), allowing its expression to follow an anterior-low to posterior-high gradient, which is crucial for specifying areal identities in the dorsal telencephalon (*Liu et al., 2000*; *Qiu et al., 1994*; *Tripodi et al., 2004*; *Armentano et al., 2006*; *Zhou et al., 2001*; *Armentano et al., 2007*). Other FGF8 downstream targets promote anterior identity by inhibiting *Nr2f1* (*O'Leary et al., 2007*; *O'Leary and Sahara, 2008*), creating a cross-talk where FGFs and *Nr2f1* exert opposite effects on the A/P cortical axis (*Bertacchi et al., 2019*). Interestingly, human *NR2F1* also exhibits a low anterior to high posterior expression gradient in the early telencephalon (*Foglio et al., 2021*; *Alzu'bi et al., 2017a*; *Clowry et al., 2018*), raising the possibility that FGFs could modulate similar downstream effector genes in the human brain. However, despite significant advances in understanding FGF-related physiology and pathology, most studies have relied on animal (often murine) models, where FGF depletion results in severe phenotypes (*Meyers et al., 1998*; *Sun et al., 1999*; *Reifers et al., 1998*; *Shanmugalingam et al., 2000*). Hence, the role of FGFs, particularly FGF8, in human progenitor cells and neurons remains poorly understood, as does the contribution of FGF8 signaling and its target gene regulation in human diseases.

Self-organizing human brain organoids offer an unprecedented tool for modeling early neural development *in vitro*, offering an ethical and practical alternative to studying early fetal brain development (*Velasco et al., 2020*; *Qian et al., 2019*; *Chiaradia and Lancaster, 2020*; *Quadrato et al., 2017*; *Lancaster et al., 2013*; *Lancaster and Knoblich, 2014*; *Renner et al., 2017*; *Bagley et al., 2017*; *Qian et al., 2020*). Previous studies utilizing 2D neural protocols (*Chambers et al., 2009*; *Smukler et al., 2006*; *Gaspard et al., 2009*), followed by adaptations for 3D organoid protocols (*Kadoshima et al., 2013*; *Qian et al., 2018*; *Qian et al., 2016*), demonstrated that pluripotent cells adopt a dorsal/anterior (*i.e.* telencephalic) neural fate when shielded from posteriorizing factors such as TGFβ, BMPs, retinoic acid, and WNTs. By building upon previous methods (*Chambers et al., 2009*; *Qian et al., 2018*; *Qian et al., 2016*), we optimized a hybrid 2D/3D *in vitro* culture system that facilitates the efficient formation of telencephalic FOXG1+ tissue and allows for the assessment of FGF8-mediated effects on neural cell differentiation and identity acquisition. Our findings indicate that FGF8 treatment leads to altered regional identities in 3D organoids compared to untreated controls, resulting in the development of multi-regional organoids with distinct co-developing brain domains. FGF8 efficiently modulates *NR2F1* and other genes associated with brain disorders in human FOXG1+ telencephalic cells, underscoring its critical role in the fine-tuning of key neurodevelopmental and pathogenic pathways. Notably, prolonged exposure to FGF8 affects not only A/P areal identity but also D/V telencephalic cell identity. As a result, FGF8-treated brain organoids exhibit an imbalance in the production of excitatory and inhibitory neurons, which impacts the formation of electrical neural networks. Collectively, our data redefine the role of FGF8 as a crucial morphogen for regional patterning and the establishment of distinct D/V and A/P telencephalic identities in human cells, thus highlighting its significance in modulating the expression of key developmental and NDD-related genes during the organization of the human brain.

## Results

### An optimized 2D/3D organoid protocol allows a fast and reproducible generation of telencephalic cells

Based on previous methods (*Chambers et al., 2009*; *Qian et al., 2018*), we optimized a human brain organoid protocol to study how FGF8 signaling impacts neuronal development and differentiation (*Figure 1A*). For fast and efficient induction of neural progenitors, we employed a dual inhibition

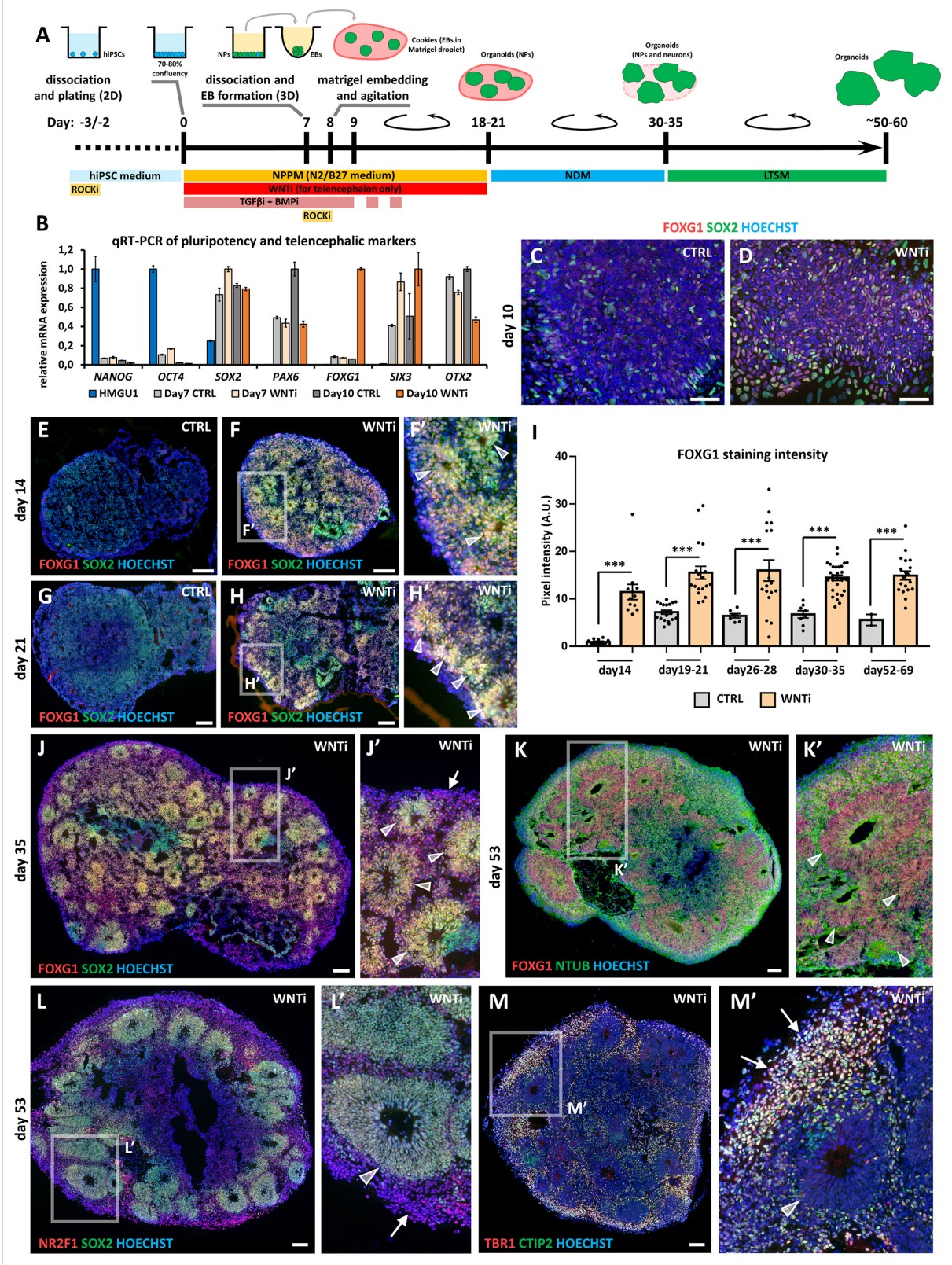

**Figure 1.** Hybrid 2D/3D protocol for fast and reproducible generation of human cortical organoids. (**A**) Schematic of the hybrid 2D/3D method for generating telencephalic/cortical human organoids in vitro, using a triple inhibition of TGFβ, BMP, and WNT pathways (SB-431542, 5 μM; LDN-193189, 0.25 μM; XAV-939, 2 μM). On day 7, cells are dissociated and re-aggregated in 96-well plates. One day later, embryoid bodies (EBs) are embedded in Matrigel droplets (10 μL per droplet containing 1–4 EBs). These droplets, termed 'cookies', are then cultured in spinning bioreactors. (**B**) Real-time

*Figure 1 continued on next page*

*Figure 1 continued*

qRT-PCR analysis quantifying pluripotency markers (*NANOG*, *OCT4*) and telencephalic neural progenitor (NP) markers (*SOX2*, *PAX6*, *FOXG1*, *SIX3*, and *OTX2*) in undifferentiated HMGU1 hiPSCs and in day7 and day10 control (CTRL) and WNT-inhibited (WNTi) samples, as indicated. n=2 culture wells per condition, pooled prior to RNA extraction. (**C,D**) Immunostaining for FOXG1 (red) and SOX2 (green) in day10 2D neural cultures under control (CTRL) conditions (**C**) or following WNT inhibition (WNTi) (**D**). (**E–I**) Immunostaining for FOXG1 (red) and SOX2 (green) in day14 (**E-F'**) and day21 (**G-H'**) organoids under CTRL or WNTi conditions, as indicated. White arrowheads in high-magnification images point to neural progenitor (NP) rosettes. The graph (**I**) shows quantification of FOXG1 pixel intensity in CTRL and WNTi samples across time points. n≥7 sections from n≥4 organoids from n=2 independent batches (except day52-69 CTRL sample, n=2 sections from 1 batch). (**J,J'**) FOXG1 (red) and SOX2 (green) immunostaining in day35 WNTi organoids. White arrowheads in high-magnification images indicate NP neural rosettes, while arrows highlight differentiating neurons surrounding the rosettes. (**K-M'**) Immunostaining for FOXG1 (red) and NTUB (green) (**K, K'**), NR2F1 (red) and SOX2 (green) (**L, L'**), and TBR1 (red) and CTIP2 (green) (**M, M'**) in day53 WNTi organoids. High-magnification images highlight FOXG1+ SOX2+ NR2F1+ NP rosettes/neuroepithelia (K-L'; indicated by white arrowheads) surrounded by TBR1+ CTIP2+ NR2F1+ differentiating cortical neurons (L'-M'; indicated by white arrows). Scale bars: 100 µm.

The online version of this article includes the following source data and figure supplement(s) for figure 1:

**Source data 1.** Expression level of pluripotency and telencephalic markers in 2D human progenitors.

**Source data 2.** FOXG1 staining intensity in CTRL and WNTi human organoids.

**Figure supplement 1.** Hybrid 2D/3D protocol for generation of telencephalic human organoids.

of SMAD signaling paradigm (*Chambers et al., 2009*) by treating confluent 2D hiPSC cultures with TGFβ and BMP inhibitors (SB-431542 and LDN-193189, respectively). As endogenous WNT factors can inhibit the acquisition of an anterior fate (*Bertacchi et al., 2015b*; *Lupo et al., 2013*), a chemical WNT inhibitor (XAV-939) was added for optimal induction of telencephalic regional identity. Around day5-6, clusters of radially organized neural progenitors (*i.e.* neural rosettes) were visible in brightfield microscopy (*Figure 1—figure supplement 1*). Consistently with the appearance of rosettes, analysis of key markers for pluripotency and neural differentiation by real-time qRT-PCR showed efficient neural induction by day7 (*Figure 1B*), with down-regulation of stemness markers *OCT4* and *NANOG* and upregulation of a molecular signature characteristic of antero-dorsal telencephalic neural progenitors (NPs; *SOX2*, *PAX6*, *SIX3* and *OTX2*). Notably, only NPs treated with XAV-939 (WNT inhibition; WNTi hereafter) efficiently upregulated the telencephalic marker *FOXG1* at day10 compared to control samples (CTRL; *Figure 1B*), as further confirmed by immunostaining (*Figure 1C and D*). To obtain 3D organoids, we dissociated day7 neural rosettes and re-aggregated early NPs into spherical aggregates (embryoid bodies, EBs; *Figure 1—figure supplement 1*), which were included 24 hr later in Matrigel droplets. For optimal nutrient and oxygen distribution, EB-containing Matrigel droplets (named 'cookies'; *Qian et al., 2018*) were cultured in miniaturized spinning bioreactors (*Qian et al., 2018*; *Qian et al., 2016*). After a few additional days of 3D culture, NPs spontaneously re-organized as multiple radially structured rosettes (*Figure 1E–F'*), but only WNTi organoids showed a significantly higher number of FOXG1+ rosettes compared to CTRL ones (*Figure 1E–I*). After 10–15 days (day35) of culture in neural differentiation medium (NDM), SOX2+ FOXG1+ NP rosettes started to be surrounded by differentiating neurons (*Figure 1J and J'*). From this stage onwards, the culture medium was supplemented with pro-survival and anti-apoptotic elements to provide optimal conditions for long-term cultures (long-term survival medium, LTSM). At around day50, FOXG1+ NR2F1+ SOX2+ NP rosettes were surrounded by differentiated neurons (*Figure 1K–M'*), which were positive for neural-Tubulin (NTUB) staining (*Figure 1K and K'*) and expressed markers of cortical layers such as TBR1 and CTIP2 (*Figure 1M and M'*). In summary, by combining previous protocols for the induction of human brain cells, we associated the high yield and rapidity of 2D neural induction with the optimized growth of 3D neural structures in spinning bioreactors, obtaining anterior NPs in 7 days and highly organized telencephalic organoids in 1 month of culture.

## FGF8 treatment modulates telencephalic target genes in brain organoids

In the developing mouse brain, distinct sources of diffusing FGF8 fine-tune the expression of several genes. In WNTi human organoids, we found that early FGF8 treatment (starting at day5; *Figure 2—figure supplement 1A and B*) reduced FOXG1 expression, in line with early FGF8 inducing posterior rather than anterior identity (*Stevens et al., 2010*). The addition of FGF8 to the culture medium starting at day10-11 (*Figure 2A*) or later at day20 did not affect FOXG1 expression (*Figure 2—figure supplement 1A and B*), indicating preservation of telencephalic identity. However, day20 FGF8

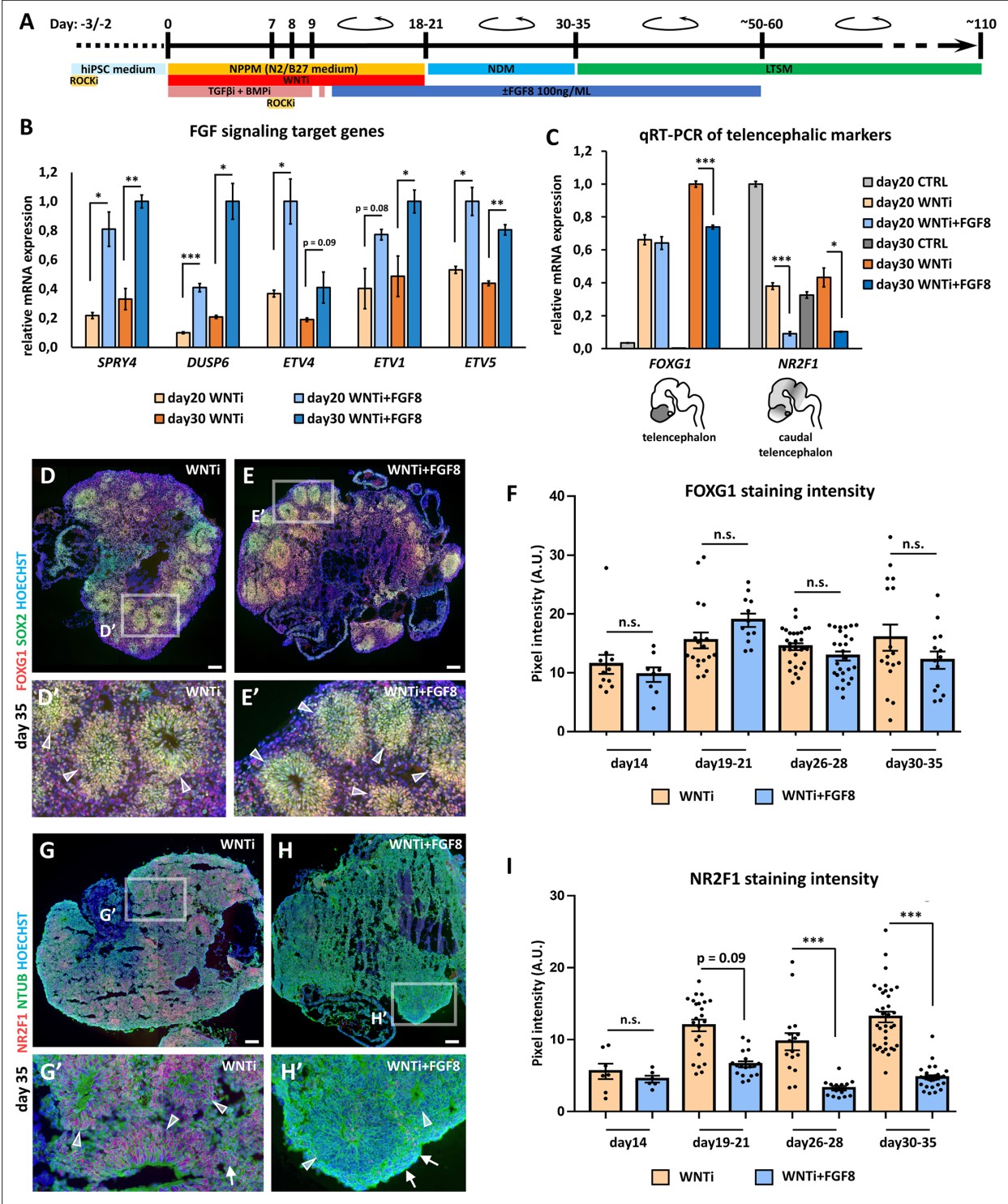

**Figure 2.** FGF8-mediated regulation of target gene expression in FOXG1+ telencephalic organoids. (**A**) Schematic of the hybrid 2D/3D method for applying FGF8 treatment on telencephalic/cortical human organoids *in vitro*. FGF8 (100 ng/mL) was added to the neural progenitor patterning medium (NPPM) beginning on day10-11 (blue bar) and maintained through subsequent culture steps until approximately day50-60. (**B**) Real-time qRT-PCR analysis of FGF8 target gene expression (*SPRY4*, *DUSP6*, *ETV4*, *ETV1*, and *ETV5*) in day20 and day30 organoids treated with WNT inhibition alone (WNTi) or in combination with FGF8 (WNTi + FGF8), as indicated. n=3 organoids per condition, pooled prior RNA extraction. (**C**) Real-time qRT-PCR quantification of *FOXG1* (telencephalic marker) and *NR2F1* (caudal telencephalic marker and FGF8 target) expression in day20 and day30 control (CTRL), WNT-inhibited (WNTi), and FGF8-treated (WNTi + FGF8) samples, as indicated. FGF8 treatment effectively downregulates *NR2F1* expression in WNTi + FGF8 organoids compared with WNTi organoids. n=3 organoids per condition, pooled prior RNA extraction. (**D–F**) Immunostaining for FOXG1

*Figure 2 continued on next page*

*Figure 2 continued*

(red) and SOX2 (green) in day35 WNTi and WNTi + FGF8 organoids, as indicated. FGF8 treatment does not significantly alter FOXG1 expression. White arrowheads in high-magnification images indicate SOX2+ NR2 F1+ NPs within rosettes. Graph (**F**) shows pixel intensity quantification of FOXG1 staining in WNTi and WNTi + FGF8 organoids at different time points. n≥8 sections from n≥4 organoids from n≥2 distinct batches. (**G–I**) NR2F1 and NTUB (red and green, respectively, in G-H') immunostainings on day35 WNTi and WNTi + FGF8 organoids, as indicated. FGF8 treatment efficiently modulates NR2F1 expression (compare G and H). High-magnification images (**G'** and **H'**) show neural rosettes (NTUB^low, indicated by white arrowheads) and differentiating neurons (NTUB^high, indicated by white arrows), both expressing NR2F1 (red) in WNTi organoids, but lacking NR2F1 in WNTi + FGF8 organoids. Graph (**I**) displays pixel intensity quantification of NR2F1 staining in WNTi and WNTi + FGF8 organoids over time. n≥6 sections from n≥4 organoids from n≥2 distinct batches. Scale bars: 100 μm.

The online version of this article includes the following source data and figure supplement(s) for figure 2:

**Source data 1.** Quantitative RT-PCR data for FGF target genes in human organoids.

**Source data 2.** Quantitative RT-PCR data for FOXG1 and NR2F1 in human organoids.

**Source data 3.** FOXG1 staining intensity in WNTi and WNTi + FGF8 human organoids.

**Source data 4.** NR2F1 staining intensity in WNTi and WNTi + FGF8 human organoids.

**Figure supplement 1.** Effect of early or late FGF8 treatment on FOXG1 and NR2F1 expression in human organoids.

**Figure supplement 1—source data 1.** FOXG1 staining intensity following early or late FGF8 treatment.

**Figure supplement 1—source data 2.** NR2F1 staining intensity following early or late FGF8 treatment.

**Figure supplement 2.** FGF8-mediated control of NR2F1 level in FOXG1+ telencephalic organoids.

**Figure supplement 2—source data 1.** FOXG1 staining intensity in WNTi and WNTi + FGF8 human organoids.

**Figure supplement 2—source data 2.** NR2F1 staining intensity in WNTi and WNTi + FGF8 human organoids.

treatment was less efficient in modulating the FGF8-target NR2F1 (*Figure 2—figure supplement 1C and D*), suggesting that earlier treatment was more appropriate for efficient FGF8 pathway modulation. Hence, we chose day10-11 as the starting time point for FGF8 treatment (hereafter referred to as WNTi + FGF8 condition), which preserved FOXG1 expression while efficiently modulating FGF8 target genes. Real-time qRT-PCR analysis of known FGF8 target genes (*SPRY4*, *DUSP6*, *ETV4*, *ETV1*, and *ETV5*) confirmed that our set-up efficiently activated FGF signaling in day20 and day30 organoids (*Figure 2B*). As a specific read-out of FGF8 treatment on the expression of telencephalic targets, we stained control (WNTi) and treated (WNTi + FGF8) organoids for NR2F1 and FOXG1 at different time points (*Figure 2C–I* and *Figure 2—figure supplement 2*). NR2F1 expression, still low on day19 but higher on day26, was efficiently modulated by FGF8 (*Figure 2—figure supplement 2A–H'*). Real-time qRT-PCR confirmed FGF8-mediated inhibition of *NR2F1* on day20 and day30 cultures (*Figure 2C*), while *FOXG1* levels were partially affected upon FGF8 treatment on day30 (*Figure 2C*). Despite this, WNTi + FGF8 organoids still largely expressed FOXG1 at later stages (day35 and day53; *Figure 2D–F* and *Figure 2—figure supplement 2I–J'*) alongside efficient modulation of NR2F1 protein levels (*Figure 2G–I* and *Figure 2—figure supplement 2K–L'*). FGF8 was maintained in the culture medium until day ~60 to prevent any increase of NR2F1 expression back to normal levels (*Figure 2—figure supplement 2M–P'*). Thus, our FGF8 culture conditions efficiently modulate NR2F1 in telencephalic organoids starting at day19-21 and until day69-74, while maintaining FOXG1 expression at high levels (*Figure 2—figure supplement 2Q and R*), validating the efficacy of FGF8 treatment *in vitro* and suggesting an evolutionary conservation of the NR2F1-FGF8 regulatory molecular axis from mice to humans.

## Single-cell RNA sequencing (scRNAseq) reveals multiple progenitor and neuronal classes present in human organoids

To investigate the transcriptomic signature of cerebral organoids upon FGF8 treatment, we employed a scRNAseq approach to compare control organoids (WNTi) with FGF8-treated telencephalic organoids (WNTi + FGF8). Two distinct batches of both WNTi and WNTi + FGF8 organoids showed high reproducibility in terms of cell cluster compositions (*Figure 3A*), allowing us to pool together individual batches to achieve a higher number of cells per cluster (n° of cells per cluster in *Figure 3—figure supplement 1A*). Bioinformatic analysis of the whole cell population (WNTi and WNTi + FGF8 cells together) identified 15 distinct cellular clusters (shown as UMAP projections; *Figure 3B*), whose composition was determined *via* the expression of well-known reference markers as a read-out

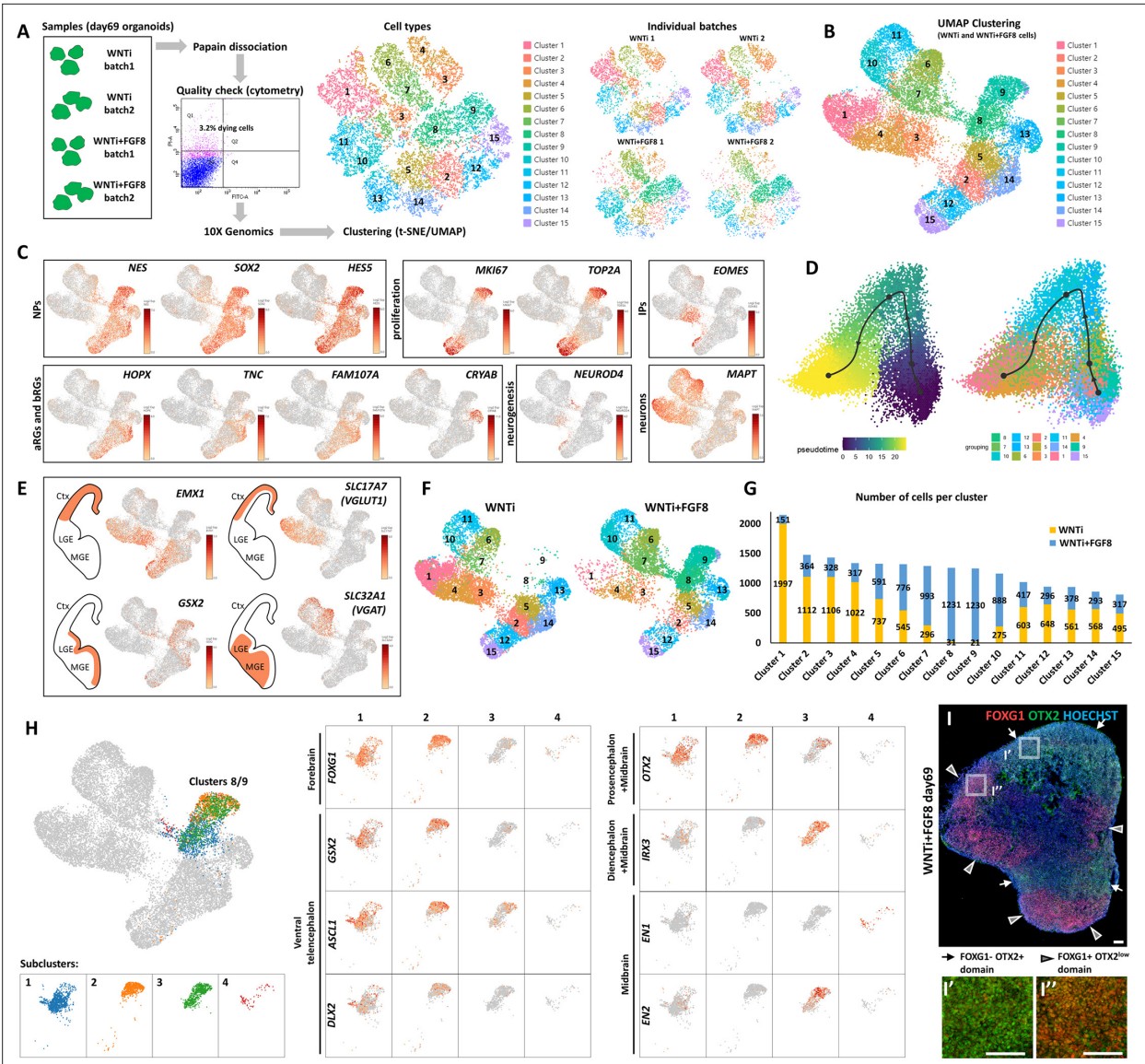

**Figure 3.** Single-cell RNA sequencing (scRNAseq) analysis of FGF8-induced cellular and molecular changes in human organoids. (**A**) Experimental setup for scRNA-seq analysis of control (WNTi) and FGF8-treated (WNTi + FGF8) telencephalic organoids at day 69. Two independent batches of WNTi and WNTi + FGF8 organoids (each containing 2–3 organoids) were dissociated into single cells and processed using Chromium (10 X Genomics technology). Cells were clustered and visualized in 2D space using t-SNE and UMAP algorithms. (**B**) UMAP clustering of WNTi and WNTi + FGF8 cells, identifying 15 distinct clusters. (**C**) Expression levels of known markers for different cell types, including neural progenitor cells (NPs: *NESTIN, SOX2, HES5*), proliferating progenitors (*KI67, TOP2A*), intermediate progenitors (IPs: *EOMES*), apical and basal radial glia cells (aRGs and bRGs: *HOPX, TNC, FAM107A, CRYAB*), and differentiating/differentiated neurons (*NEUROD4* and *MAPT*, respectively). (**D**) Trajectory analysis showing the most probable developmental progression from NP clusters (2, 5, 8, 9, 12, 15) to post-mitotic cell types (notably clusters 1, 3, 4, 6, 7). (**E**) Expression level and cluster distribution of dorsal glutamatergic markers *EMX1* and *SLC17A7* (also called *VGLUT1*) and ventral GABAergic markers *GSX2* and *SLC32A1* (also called *VGAT*), indicating the coexistence of both glutamatergic and GABAergic NPs and neurons within FOXG1+ telencephalic organoids. (**F, G**) UMAP clustering of WNTi and WNTi + FGF8 cells shown separately, illustrating 15 distinct clusters and their respective proportions in each condition. Panel (**G**) shows the number of cells in each cluster originating from WNTi (yellow) or WNTi + FGF8 (blue) organoids. (**H**) UMAP projection of day69 organoid scRNA-seq data, identifying four cellular groups through sub-clustering analysis on WNTi + FGF8 clusters 8 and 9. Center and right panels display expression levels of markers for the forebrain (*FOXG1*), ventral telencephalon (*GSX2, ASCL1* and *DLX2*), forebrain/midbrain (*OTX2*), diencephalon/mesencephalon (*IRX3*), and mesencephalon (*EN1, EN2*) across the four sub-clusters. (**I–I''**) Immunostaining for FOXG1 (red) and OTX2 (green) in day69 WNTi + FGF8 organoids, showing distinct FOXG1+ and FOXG1- regions. White arrows indicate FOXG1- OTX2+ non-telencephalic areas (high magnification in I'), while arrowheads denote FOXG1+ OTX2$^{low}$ telencephalic areas (high magnification in I''). Ctx, cortex; MGE, medial ganglionic eminence; LGE, lateral ganglionic eminence.

The online version of this article includes the following source data and figure supplement(s) for figure 3:

*Figure 3 continued on next page*

*Figure 3 continued*

**Source data 1.** Trajectory analysis for all cell clusters.

**Source data 2.** Cell counts per cluster by origin (control WNTi or treated WNTi + FGF8 organoids).

**Figure supplement 1.** Single-cell RNA sequencing (scRNAseq) and VoxHunt similarity map analysis of 2-month-old telencephalic organoids.

**Figure supplement 1—source data 1.** Cell count per cluster from scRNAseq analysis of day 69 organoids.

**Figure supplement 1—source data 2.** VoxHunt analysis report.

**Figure supplement 1—source data 3.** Trajectory analysis of progenitor cell clusters.

**Figure supplement 1—source data 4.** Trajectory analysis of glutamatergic clusters.

**Figure supplement 1—source data 5.** Trajectory analysis of GABAergic clusters.

**Figure supplement 2.** FGF8-dependent induction of diencephalic and mesencephalic markers in telencephalic organoids.

**Figure supplement 2—source data 1.** Number of *FOXG1*+ cells per cluster.

**Figure supplement 2—source data 2.** Quantitative RT-PCR data for telencephalic and mesencephalic markers in human organoids.

**Figure supplement 2—source data 3.** Number of cells expressing distinct positional markers in clusters 8 and 9.

**Figure supplement 3.** FGF8-dependent induction of OTX2+ domains in multi-regional organoids.

**Figure supplement 3—source data 1.** OTX2+ area percentages in multi-regional organoids.

---

of cell identity (*Figure 3C* and *Figure 3—figure supplement 1B*). Cell clusters comprised NPs (*NESTIN*+, *SOX2*+, and *HES5*+ cells in clusters 2, 5, 8, 9, 12, 13, 14, and 15), some of which were actively proliferating (*KI67*+ and *TOP2A*+), and neurons (*MAPT*+ cells in clusters 1, 3, 4, 6, 7, 10, and 11). Expression of neurogenic (*NEUROD4* and *NEUROG1*) and post-mitotic neuronal markers (*DCX*, *RBFOX3*, and *MAPT*) highlighted the existence of both differentiating and differentiated neurons (*Figure 3C* and *Figure 3—figure supplement 1B*). *EOMES* (also known as *TBR2*) expression identified intermediate progenitors, while expression of *HOPX*, *TNC*, and *FAM107A* indicated both apical and basal radial glia (*Pollen et al., 2015*). Interestingly, a *bona fide* marker for late truncated radial glia (*CRYAB*; *Nowakowski et al., 2016*), which is normally expressed in the neocortex *in vivo* starting at 14.5 post-conceptional weeks (PCW), was specifically expressed in NP cluster 13 (*Figure 3C*). We reasoned that multiple NP and neuronal types co-existed in telencephalic organoids, and trajectory analysis confirmed that NP clusters (clusters 2, 5, 8, 9, 12, and 15) gradually converted into post-mitotic neurons (clusters 1, 3, 4, 6, and 7; *Figure 3D*), suggesting that key steps of human brain development were recapitulated *in vitro*.

To assign the identities of distinct cell clusters in a more unbiased way, we used the VoxHunt spatial brain mapping and heatmap of similarity score tools (*Fleck et al., 2021*; *Figure 3—figure supplement 1C and D*), which evaluate the similarity between the expression profile of each cluster and those of spatial and single-cell transcriptome reference datasets. We established a high resemblance of most clusters to the dorsal telencephalon (pallium), with clusters 2, 5, 12, 14, and 15 showing high similarity scores with progenitors, while clusters 1, 3, and 4 included cortical neurons (*Figure 3—figure supplement 1C and D*). However, some ventral telencephalic (subpallium) clusters were also present (clusters 6 and 7), consistent with previous reports showing that ventral GABAergic identity can spontaneously arise in dorsally patterned organoids (*Velasco et al., 2019*). Notably, clusters 8, 9, 10, and 11 showed a mixed identity, as they scored high for both dorsal and ventral telencephalon characteristics and/or for more posterior brain regions (*Figure 3—figure supplement 1D*). Analysis of known glutamatergic and GABAergic markers substantiated the D/V identity of organoid NPs and neurons, which could be subdivided into two pools of glutamatergic (*EMX1*+ and *SLC17A7*+) and GABAergic (*GSX2*+ and *SLC32A1*+) cells (*Figure 3E*). Additionally, trajectory analysis of different clusters highlighted a dynamic state of developmental trajectories (*Figure 3—figure supplement 1E*). Together, our 2 months-old organoid model displays a transcriptomics signature of mixed dorsal (glutamatergic) and ventral (GABAergic) cellular identities, reflecting the cellular and molecular properties of a 14.5 PCW human neocortex.

## FGF8 treatment increases brain regional heterogeneity

While widespread *FOXG1* expression supported the general telencephalic identity of both WNTi and WNTi + FGF8 treated organoids (*Figure 3—figure supplement 2A and B*), qRT-PCR showed a partial reduction of the anterior marker *SIX3*, induction of *OTX2* a telencephalic marker at early

stages (*Ostermann et al., 2019*) before becoming restricted to more posterior regions at later stages (*Puelles et al., 2004*), and induction of the mesencephalic marker *EN2* in WNTi + FGF8 organoids (*Figure 3—figure supplement 2C*), suggesting that long-term FGF8 treatment might induce the formation of regions other than the forebrain. Hence, we explored the expression of key markers of A/P regional identity by focusing on clusters 8 and 9, as these cellular populations were almost absent in control (WNTi) organoids and only appeared in treated (WNTi + FGF8) samples (*Figure 3F and G* and *Figure 3—figure supplement 2D and E*). Although cluster 8/9 cells were largely positive for telencephalic markers such as *FOXG1* (50% positive cells) and *SFRP1* (70% positive cells) (*Figure 3—figure supplement 2A–E*), they also displayed expression of the diencephalic gene *SIX3* (>40% positive cells) and mesencephalic markers *OTX2* (>30% positive cells), *IRX3* (20% positive cells), and *EN2* (10% positive cells) upon FGF8-treatment (*Figure 3—figure supplement 2D and E*). These data indicate that clusters 8 and 9, which are only present in FGF8-treated organoids, are mainly composed of telencephalic progenitors (*FOXG1+*, *SFRP1+*, *GAS1+*, and *FZD8+*) but also contain some diencephalic (*SIX3+*) and mesencephalic (*IRX3+*, *OTX2+*, *EN2+*) cells (*Figure 3—figure supplement 2E*), suggesting concomitant FGF8-driven induction of non-telencephalic regional identities.

To further distinguish cell types in scRNAseq clusters 8 and 9, we performed a sub-clustering analysis, which detected four main cellular groups (*Figure 3H*). While two sub-clusters expressed *FOXG1* together with ventral telencephalic markers (*GSX2, DLX2* and *ASCL1*), the remaining two were negative for *FOXG1* but positive for *OTX2, IRX3, EN1* and/or *EN2* (*Figure 3H*). To directly visualize the co-existence of distinct organoid domains, we performed double staining for FOXG1 (telencephalon) and OTX2 (diencephalon/mesencephalon) on treated (WNTi + FGF8) organoids (*Figure 3I–I″*) and confirmed that FOXG1-negative domains were indeed positive for OTX2, most probably corresponding to the diencephalic/mesencephalic clusters 8 and 9 identified in scRNAseq data. Quantification of OTX2 expression revealed that a variable portion of the organoid mass (approximately 15%, reaching a maximum of ~30% in some organoids) is occupied by OTX2+ FOXG1- non-telencephalic domains in late-cultured multi-regional organoids (*Figure 3—figure supplement 3*). Collectively, our data suggest that while WNTi organoids develop as uniform FOXG1+ telencephalic organoids predominantly expressing cortical markers, FGF8-treated organoids form distinct and segregated regional domains. Therefore, elevated levels of FGF8 can enhance the complexity of human cultured organoids *in vitro* by promoting the formation of multi-regional structures, where distinct brain regions co-exist and develop concurrently within a single aggregate.

## FGF8 alters dorso/ventral cell specification of telencephalic domains

While the expression of diencephalic/mesencephalic markers was limited to clusters 8 and 9 only (*Figure 3—figure supplement 2D and E*), most of the remaining clusters expressed high levels of *FOXG1* (*Figure 3—figure supplement 2A and B*). Therefore, we focused on the effect of FGF8 treatment on these more abundant *FOXG1*-expressing telencephalic populations. Comparison of control (WNTi) and treated (WNTi + FGF8) samples in terms of cell abundance per cluster revealed an increased cell count in clusters 6 and 7 -representing cells of ventral identity- in response to FGF8 treatment (*Figure 3F and G*). In contrast, clusters associated with glutamatergic NPs (clusters 2, 5, 12, 14, and 15) and neurons (clusters 1, 3, and 4) were more abundantly populated in control organoids but less represented in FGF8-treated ones (*Figure 3F and G*). The differing abundance of cell clusters in control and treated organoids suggested a change in cellular composition following FGF8 treatment, prompting us to explore the expression of key dorsal glutamatergic and ventral GABAergic markers on WNTi and WNTi + FGF8 UMAP projections (*Figure 4A and B*). Glutamatergic NP and neuronal markers such as *EMX1, NEUROD6, NEUROD2, TBR1, SOX5, BCL11B, LHX2, NEUROG2, NF1A,* and *SLC17A7* were highly reduced in FGF8-treated organoids (*Figure 4A* and *Figure 4—figure supplement 1A*), consistent with a lower cell number in clusters 1, 3, and 4. Notably, the upper cortical layer marker *SATB2* was completely absent in WNTi + FGF8 cluster 1 (*Figure 4—figure supplement 1A*). Conversely, ventral GABAergic markers such as *ASCL1, DLX1, DLX2, PBX3, GAD1,* and *GAD2* were increased in clusters 6, 7, 8, and 9 (*Figure 4B* and *Figure 4—figure supplement 1B*), supporting a ventralization of FGF8-treated samples compared to non-treated ones. An unbiased VoxHunt analysis of the correlation score with brain areas confirmed that FGF8 treatment decreased dorsal (pallial) while increasing ventral (subpallial) telencephalic properties (*Figure 4—figure supplement 1C and D*). Interestingly, ventral medial ganglionic eminence (MGE) transcripts (*SHH, LHX8,* and

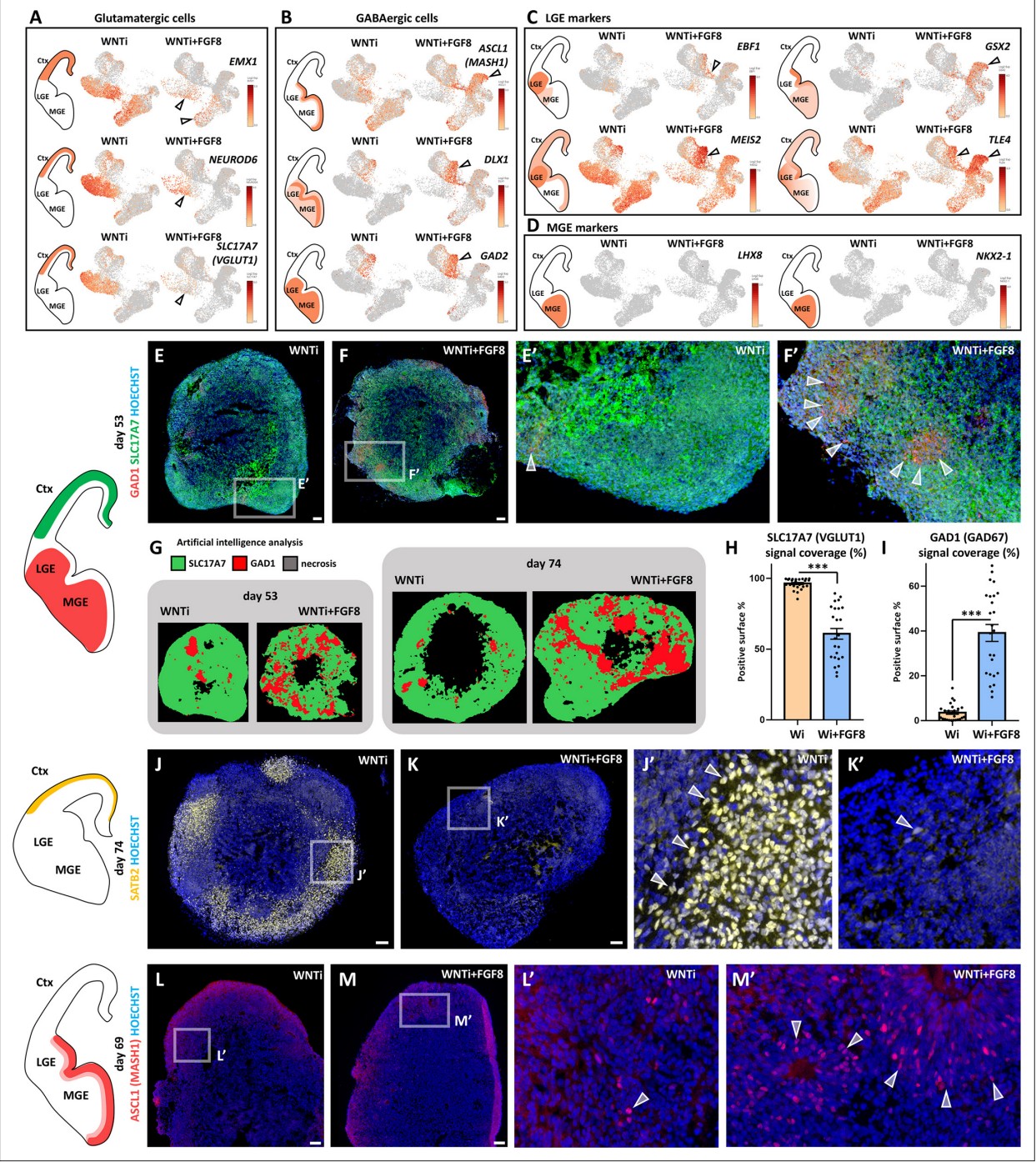

**Figure 4.** Effects of FGF8 treatment on the dorso-ventral cellular composition of telencephalic organoids. (**A, B**) Expression levels of markers identifying dorsal glutamatergic NPs and neurons (Panel A; *EMX1, NEUROD6, SLC17A7*) and ventral GABAergic NPs and neurons (Panel B; *ASCL1, DLX1, GAD2*). Black arrowheads highlight clusters with the most notable changes in marker and cell abundance after FGF8 treatment. (**C,D**) Expression levels of known markers identifying ventral GABAergic NPs and neurons in the lateral ganglionic eminence (LGE) (Panel C; *EBF1, GSX2, MEIS2* and *TLE4*) and the medial ganglionic eminence (MGE) (Panel D; *NKX2-1* and *LHX8*). Black arrowheads in C highlight clusters with the largest differences in LGE marker and cell abundance following FGF8 treatment. (**E-F'**) GAD1 (red) and SLC17A7 (green) immunostaining in day53 control (WNTi) and treated (WNTi + FGF8) organoids, as indicated. The distribution of these markers *in vivo* is shown in the brain scheme on the left. (**G–I**) HALO software artificial intelligence (AI) analysis of marker distribution in day53 and day74 organoids. Representative images display areas automatically identified as SLC17A7+ (green), GAD1+ (red), and necrotic (black). Graphs in H and I show the proportions of SLC17A7+ (**H**) and GAD1+ (**I**) surface areas in day 74 organoid sections, as quantified by HALO AI; n≥8 sections from n≥4 organoids from n=1 batch per time point. (**J-K'**) Immunostaining for SATB2 (yellow) in day74 WNTi and WNTi + FGF8 organoids, as indicated. The left schematic depicts the *in vivo* distribution of SATB2+ neurons, and ***Figure 4—figure supplement 2***

*Figure 4 continued on next page*

*Figure 4 continued*

quantifies SATB2+ neuron density in these and additional samples. (**L-M'**) Immunostaining for ASCL1 (red) in day69 WNTi and WNTi + FGF8 organoids, as indicated. Left schematic shows the *in vivo* distribution of ASCL1+ ventral progenitors, with cell density detailed in *Figure 4—figure supplement 2*. Scale bars: 100 μm. Ctx, cortex; MGE, medial ganglionic eminence; LGE, lateral ganglionic eminence.

The online version of this article includes the following source data and figure supplement(s) for figure 4:

**Source data 1.** SLC17A7 (VGLUT1)-positive area in human organoids.

**Source data 2.** GAD1 (GAD67)-positive area in human organoids.

**Figure supplement 1.** Changes in cellular composition and glutamatergic/GABAergic identity upon FGF8 treatment.

**Figure supplement 1—source data 1.** VoxHunt analysis report.

**Figure supplement 2.** Immunostaining analysis of glutamatergic and GABAergic cellular composition of control and FGF8-treated telencephalic organoids.

**Figure supplement 2—source data 1.** GAD1 (GAD67) pixel intensity in human organoids.

**Figure supplement 2—source data 2.** TBR1 and CTIP2 co-expression in human organoids.

**Figure supplement 2—source data 3.** SATB2 cell abundance in human organoids.

**Figure supplement 2—source data 4.** ASCL1 (MASH1) cell abundance in human organoids.

**Figure supplement 3.** Regional identity-dependent effect of FGF8 treatment on D/V identity and target gene modulation.

**Figure supplement 3—source data 1.** Percentage of NR2F1+ cells per cluster in WNTi and WNTi + FGF8 organoids.

---

*NKX2-1*) were not induced in WNTi + FGF8 organoids, whereas lateral ganglionic eminence (LGE)-enriched ones (*EBF1, GSX2,* and *PBX3*), particularly those associated with striatal fate (*MEIS2,* and *TLE4*), were expressed at higher levels than in control samples (*Figure 4C and D* and *Figure 4—figure supplement 1B*), suggesting that addition of FGF8 promotes a ventral LGE but not an MGE identity. Altogether, these data support an FGF8-mediated effect on the acquisition of a ventral LGE-like GABAergic identity at the expense of a dorsal glutamatergic one in telencephalic organoids.

To directly visualize a change in D/V identity in telencephalic organoids, we performed immunostainings for key GABAergic (GAD1 and ASCL1) and glutamatergic (SLC17A7, SLC17A6, and SATB2) markers (*Figure 4E–M'* and *Figure 4—figure supplement 2*). Consistent with scRNAseq data, FGF8-treated organoids showed increased GAD1 levels at the expense of SLC17A7 and SLC17A6 protein expression (*Figure 4E–I* and *Figure 4—figure supplement 2A–F*). Notably, as SLC17A6 is also expressed in non-telencephalic brain regions (*Fremeau et al., 2004*), we verified its dorsal glutamatergic identity by co-staining with the cortical markers TBR1, CTIP2, and SATB2 in control organoids (*Figure 4—figure supplement 2G–H'*). Artificial intelligence analysis using HALO software estimated a ~40% coverage of GAD1+ tissue in FGF8-treated organoids *versus* a 1.8% coverage in control ones (*Figure 4G–I*). Among key dorsal telencephalic proteins, double TBR1+ CTIP2+ cells (deep layer cortical neurons) decreased upon FGF8 treatment (*Figure 4—figure supplement 2I–M'*), whereas CTIP2+ TBR1- cells were still largely present in FGF8-treated organoids. Since these CTIP2+ cells also expressed GAD1 (*Figure 4—figure supplement 2N*), we classified them as GABAergic interneurons, similar to what was previously reported in mice (*Nikouei et al., 2016*). Furthermore, expression of the cortical marker SATB2 was highly downregulated in WNTi + FGF8 organoids at both the transcript and protein levels (*Figure 4J–K'*, *Figure 4—figure supplement 1A* and *Figure 4—figure supplement 2O–Q*), consistent with a partial loss of dorsal glutamatergic identity in favor of a ventral GABAergic identity. Among the ventral markers evaluated in scRNAseq, we detected an increased number of ASCL1+ ventral progenitors in WNTi + FGF8 organoids compared to WNTi organoids (*Figure 4L–M'* and *Figure 4—figure supplement 2R–T*). Notably, FGF8-mediated induction of ventral GAD1 and ASCL1 markers, along with the concomitant reduction of dorsal SLC17A6, TBR1, and SATB2 ones, was observed only in FOXG1+ telencephalic and not in non-telencephalic OTX2+ regions of multi-regional organoids, as supported by comparison of sequential cryostat sections of control and treated organoids (*Figure 4—figure supplement 3A–J''*). Furthermore, we found that FGF8-mediated downregulation of *NR2F1* was present only in FOXG1+ telencephalic regions, whereas *NR2F1* was not modulated by FGF8 in non-telencephalic OTX2+ regions corresponding to clusters 8 and 9 (*Figure 4—figure supplement 3G–G'', K, L*). This indicates that FGF8-mediated target gene modulation follows distinct genetic rules depending on the regional identity acquired by NPs and neurons (*Figure 4—figure supplement 3M*). Together, our results indicate a robust, telencephalon-specific effect of FGF8 on

determining the dorsal/ventral (D/V) identity—glutamatergic *versus* GABAergic—of neural progenitors and neurons during human brain development *in vitro*.

## FGF8 treatment results in altered neural network activity

To assess whether changes in D/V neuronal composition would affect functional neuroelectric activity, we used a multi-electrode array (MEA) system on 4- and 7-month-old FGF8-treated organoids (*Figure 5*). Striking differences in spontaneous activity were detected in WNTi + FGF8 organoids compared to non-treated ones (*Figure 5*; representative spike traces in *Figure 5—figure supplement 1A and B*). Notably, WNTi + FGF8 organoids showed lower spike frequency (firing rate) and decreased spike amplitude, indicating reduced electrical activity compared to WNTi organoids (*Figure 5A–C*). The network analysis, which assesses the synchronicity of spike events and serves as a read-out of neural network formation (*Trujillo et al., 2019*), also highlighted differences between the two types of organoids (*Figure 5D–F*). While WNTi organoids displayed a consistent degree of network formation with a high percentage of spikes occurring within bursts (indicating a well-organized and synchronous network activity), WNTi + FGF8 organoids showed lower synchronicity, with a higher percentage of random spikes falling outside of bursts. Analysis of burst metrics also revealed a higher number of spikes per burst and peak firing rates in control organoids (*Figure 5D and F*), whereas FGF8-treated organoids displayed lower activity levels in terms of the number and frequency of spikes per burst (*Figure 5E and F*). Nevertheless, synchronous events were still detected in WNTi + FGF8 organoids, and the average burst frequency remained unchanged (*Figure 5E and F*), suggesting a good level of spontaneous circuitry organization and maturation despite lower activity levels. Furthermore, axonal tracking identified efficient signal transduction along tracts originating from WNTi organoids, which expanded across the MEA electrodes with high conduction velocity and high latency of signal propagation (*Figure 5G–I*). In contrast, FGF8-treated organoids generated lower amplitude signals at the initiation site, which struggled to propagate over long distances (*Figure 5H and I*), suggesting intrinsic inhibition that reduced the propagation of spontaneous signals. Axonal tracking analysis performed at different maturation time points also showed that most differences between control and treated organoids remained stable over time (*Figure 5—figure supplement 1C–G*). Given that low spike frequency, low spike amplitudes, and a high percentage of random activity outside synchronous events are consistent with the inhibitory activity of GABAergic neurons (*Mossink et al., 2022*), we performed double immunostaining for SLC17A6 and GAD1 on organoids detached from MEA chips. This allowed us to correlate our functional recordings with the cellular and molecular identity of recorded neurons, confirming an increased number of GABAergic neurons in FGF8-treated samples (*Figure 5J*). Finally, to challenge the role of GABAergic inhibitory neurons in complex 3D organoid circuits, we tested the effects of transient GABA-A receptor inactivation on WNTi + FGF8 organoids *via* Bicuculline treatment (*Figure 5—figure supplement 2*). In the presence of 10 µM Bicuculline, WNTi + FGF8 organoids showed increased spike amplitudes (*Figure 5—figure supplement 2A–C*) along with a significant increase in spikes occurring within bursts (*Figure 5—figure supplement 2D–F*), suggesting that GABA release contributes to the lower spike amplitude and decreased network synchronicity in 3D organoids. These data strongly indicate that prolonged FGF8 exposure affects neuronal identity and function (in terms of glutamatergic *versus* GABAergic balance and activity), consequently influencing the spontaneous electrical activity of neuronal circuits developing in 3D organoids.

## FGF8 alters dorso/ventral specification of glutamatergic and GABAergic populations

As the abundance of cells in specific clusters was highly impacted upon FGF8 treatment, we reasoned that the modulation of specific D/V genes (*Figure 4*) could result from two overlapping factors: (i) a varying number of cells per cluster and (ii) a differential expression level of D/V genes within a given cellular population. To disentangle these parameters, we performed an analysis of differentially expressed genes (DEGs) on specific cell populations (clusters) by comparing WNTi to WNTi + FGF8 organoids, visualizing the most strongly up- or down-regulated genes in volcano-plots (*Figure 6*). This analysis allows for normalization of gene expression levels to minimize potential bias from differences in cell abundance between control and treated organoids.

By comparing WNTi and WNTI + FGF8 samples on proliferating (clusters 12 and 15) and non-proliferating (clusters 2 and 5) glutamatergic progenitors, *NR2F1* emerged as the most strongly

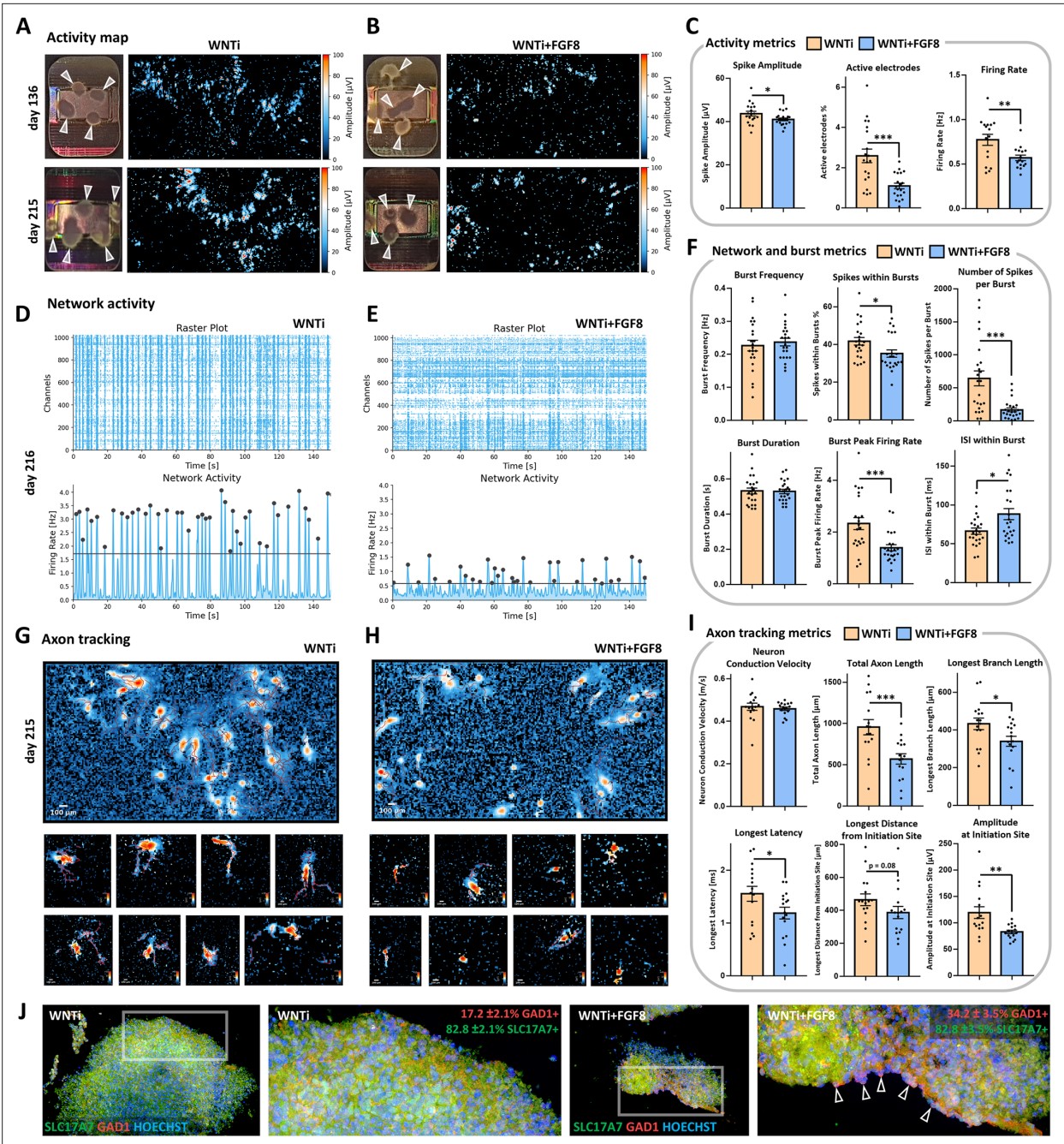

**Figure 5.** Electrophysiological spontaneous activity of control and FGF8-treated 3D organoid networks. (**A–C**) Spontaneous activity maps showing mean spike amplitude in 4-month-old (day136; upper row) and 7-month-old (day215; lower row) control (WNTi; **A**) and treated organoids (WNTi + FGF8; **B**), 2–3 weeks post-plating on high-density MEA chips. Mean activity metrics from three independent batches are summarized in (**C**), showing a general reduction in spike amplitude, percentage of active electrodes, and firing rate in FGF8-treated organoids. White arrowheads in electrode images indicate organoids fully or partially adhered to the MEA recording surface; note that high activity levels are observed at the organoid edges, where axonal tracts extend. (**D–F**) Temporal raster plots displaying firing events recorded by the 1024 most active electrodes (upper graphs) and their synchronicity indicative of network activity (lower graphs) in 7-month-old controls (WNTi; **D**) and FGF8-treated organoids (WNTi + FGF8; **E**), 3 weeks post-plating. Note that the instrument automatically sets the detection threshold (black bar) at varying levels, based on the average baseline activity specific to each sample (see Materials and methods). Graphs in (**F**) show network metrics (Burst frequency; Spikes within bursts) and burst metrics (number of spike per burst, burst peak firing rate, burst duration, inter-spike interval within bursts). (**G–I**) Overview fields of axon tracts (upper row) and representative images of individual neuronal tracts (lower rows) extending from organoids on the MEA surface, as detected using the automatic axon tracking function in WNTi (**G**) and WNTi + FGF8 (**H**) samples at day215. Graphs in (**I**) present axon tracking metrics (network conduction velocity, total axon or branch lengths, longest signal latency, maximum distance from initiation site, and signal amplitude), showing reduced signal amplitude and spatial propagation

*Figure 5 continued on next page*

*Figure 5 continued*

in FGF8-treated organoids. Additional axon tract images at different stages of differentiation are provided in *Figure 5—figure supplement 1*. (**J**) Immunostaining for GAD1 (red) and SLC17A6 (green) in day 145 control (WNTi) and treated (WNTi + FGF8) organoids, following detachment from MEA chips post-recording. Percentages of GAD1- and SLC17A7-positive tissue in organoids are indicated. For all graphs: data represent n=3 distinct batches (each with 2–4 organoids on the MEA chip, see Materials and methods).

The online version of this article includes the following source data and figure supplement(s) for figure 5:

**Source data 1.** Analysis of spontaneous network activity in WNTi and WNTi + FGF8 organoids on MEAs.

**Figure supplement 1.** Representative spike traces and axonal maturation of WNTi and WNTi + FGF8 organoids on HD-MEA.

**Figure supplement 1—source data 1.** Axon tracking metrics in WNTi and WNTi + FGF8 organoids on MEAs.

**Figure supplement 2.** Bicuculline-mediated inhibition of GABA-A receptors and its effect on WNTi + FGF8 organoid spontaneous activity.

**Figure supplement 2—source data 1.** Activity scan and network analysis of WNTi + FGF8 organoids on MEAs with or without Bicuculline treatment.

down-regulated gene upon FGF8 treatment, both in terms of fold change and p-value (*Figure 6A*). Another down-regulated gene in glutamatergic progenitors was *FGFR3*, which, together with *NR2F1*, is a primary target of FGF8 signaling and a key effector in regulating neocortical areal identity (*Garel et al., 2003*; *Inglis-Broadgate et al., 2005*; *Mott et al., 2010*). Gene ontology (GO) enrichment analysis showed that many DEGs were related to cell proliferation (DNA replication) or differentiation (nervous system development; cell differentiation) categories (*Figure 6—figure supplement 1*). Among FGF8-induced genes, *ZIC1* and *ZIC3* (*Figure 6A and B*) are known to maintain neural precursor cells in an undifferentiated state in the mouse medial telencephalon (*Inoue et al., 2007*). Most importantly, genes typically expressed only in ventral progenitors, such as *DLX2* and *GSX2*, were significantly upregulated in dorsal glutamatergic progenitor clusters (*Figure 6A and C*), suggesting a misspecification of glutamatergic progenitors towards a ventral identity. Similarly, post-mitotic differentiating (cluster 3) and differentiated (clusters 1 and 4) glutamatergic neurons (*Figure 6D*) showed a clear imbalance in D/V gene expression, including reduced glutamatergic markers (*NF1B, NEUROD6* and *SOX5*) and induction of *DLX2*, confirming that FGF8 can influence the establishment of a glutamatergic molecular network by inducing ectopic GABAergic marker expression. DEGs in clusters 1, 3, and 4 were generally associated with neural differentiation GO categories (axonogenesis; neurogenesis; neuron differentiation; axon guidance; cell differentiation, among others; *Figure 6—figure supplement 1*), consistent with a post-mitotic neural identity of these clusters. Among FGF8-induced genes in glutamatergic neurons, we identified *ETV1* (*Cholfin and Rubenstein, 2008*), a known target serving as an internal control for FGF8 efficiency, and again, *NR2F1* as the most significantly down-regulated gene (*Figure 6D*). Additionally, other factors modulated by FGF8 in both NPs and neurons included *AUTS2, NFIB, ZIC1, ZIC3, FOXP2, CCND2, MEIS2*, and *SOX5*, which are known to be mutated in NDDs (*Figure 6A and D*). Finally, we focused on DEG changes in ventral GABAergic cells (clusters 6 and 7; *Figure 6E*). Gene enrichment analysis revealed GO categories associated with neuronal differentiation and migration (forebrain development; CNS development; neuron migration; nervous system development, among others; *Figure 6—figure supplement 1*). FGF8 treatment affected the expression levels of *NR2F1, NTRX2, TUBB3, ZIC1, ZIC2*, and *MEIS2*, and notably, the *NR2F1* homolog *NR2F2* emerged as a highly modulated FGF8 target gene (*Figure 6E*). Together, our DEG analyses suggest FGF8-dependent misspecification of glutamatergic and GABAergic progenitors and neurons, along with the dysregulation of several genes related to normal and pathological brain development.

## FGF8 treatment modulates A/P neocortical identity and areal-specific factor expression

As the DEG analysis revealed downregulation of posterior factors such as *NR2F1* and *FGFR3* and upregulation of anterior genes *ZIC1* and *ZIC3* following FGF8 treatment, we hypothesized that FGF8 might also modulate the A/P identity of telencephalic cells, in addition to its effects on D/V identity. Notably, *NR2F1* is positioned at the top of a regulatory hierarchy controlling other region-specific factors, including *EMX2, FGFR3, SP8*, and *PAX6* (*Bertacchi et al., 2019*; *Figure 7—figure supplement 1A*). VoxHunt Similarity brain maps indicated that FGF8-treated organoids maintained a high similarity score with the anterior-most sections of the dorsal telencephalon, while similarity to posterior regions was nearly lost (*Figure 7—figure supplement 1B*). This suggests that FGF8 treatment preferentially supports the expression of anterior cortical genes. To further investigate the impact of

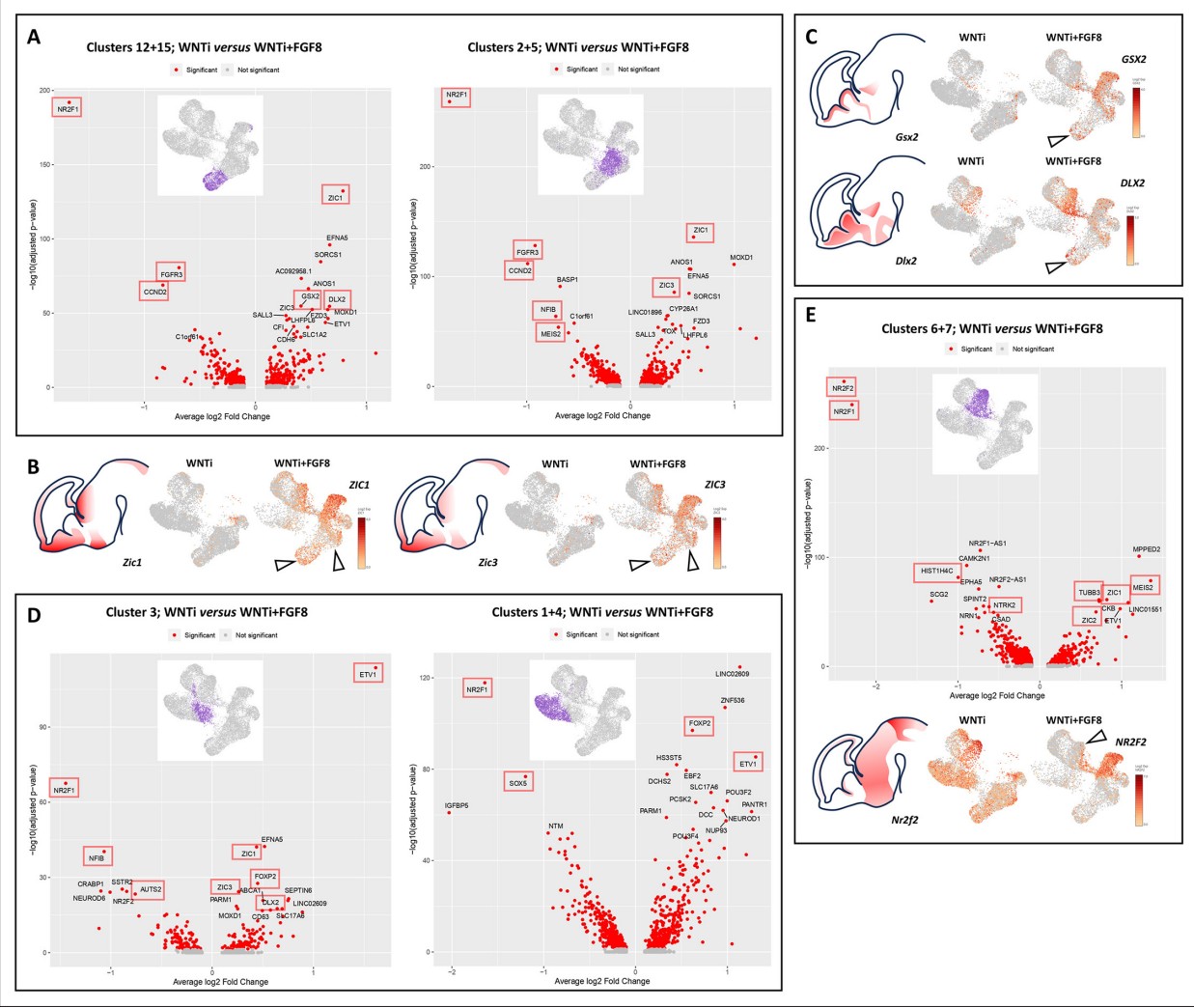

**Figure 6.** Analysis of differentially expressed genes (DEGs) in glutamatergic and GABAergic progenitors and neurons upon FGF8 treatment. (**A**) DEG analysis comparing control (WNTi) and treated (WNTi + FGF8) organoids, highlighting the most strongly (x-axis, average log2 fold change) and most significantly (y-axis, adjusted p-value in -log10) differentially expressed genes (DEGs) in clusters 12/15 (proliferating glutamatergic progenitors; left volcano plot) or clusters 2/5 (non-proliferating glutamatergic progenitors; right volcano plot). Red dots indicate significantly (adjusted P-value <0.05) regulated genes, with the names of the 20 most significant ones displayed (see source data material for a complete list of DEGs). (**B**) Expression level of *ZIC1* (left) and *ZIC3* (right) in UMAP projections of WNTi and WNTi + FGF8 samples, as indicated. Black arrowheads point to increased *ZIC1* and *ZIC3* expression in glutamatergic progenitor clusters following FGF8 treatment. (**C**) Expression level of *GSX2* (upper panel) and *DLX2* (lower panel) in UMAP projections of WNTi or WNTi + FGF8 samples, as indicated. Black arrowheads indicate increased *GSX2* and *DLX2* expression in proliferating glutamatergic progenitors upon FGF8 treatment. (**D**) DEG analysis comparing control (WNTi) and treated (WNTi + FGF8) organoids, highlighting the most significantly regulated genes in cluster 3 (early differentiating glutamatergic neurons; left volcano plot) or clusters 1/4 (differentiated glutamatergic neurons; right volcano plot). (**E**) DEG analysis comparing control (WNTi) and treated (WNTi + FGF8) organoids, showing the most significantly regulated genes in clusters 6/7 (volcano plot), corresponding to GABAergic neurons. The panel below shows the expression level of *NR2F2* in UMAP projections of WNTi and WNTi + FGF8 samples, as indicated. The black arrowhead points to decreased *NR2F2* expression in GABAergic cells following FGF8 treatment. Red boxes highlight FGF target genes or genes noted in OMIM as disease-related. Brain schematics with gene expression patterns are based on embryonic day 13.5 stainings from the Mouse Allen Brain Atlas.

The online version of this article includes the following source data and figure supplement(s) for figure 6:

**Source data 1.** Analysis of differentially expressed genes in cycling glutamatergic progenitors following FGF8 treatment.

**Source data 2.** Analysis of differentially expressed genes in non-cycling glutamatergic progenitors following FGF8 treatment.

**Source data 3.** Analysis of differentially expressed genes in early differentiating glutamatergic neurons following FGF8 treatment.

**Source data 4.** Analysis of differentially expressed genes in post-mitotic glutamatergic neurons following FGF8 treatment.

**Source data 5.** Analysis of differentially expressed genes in GABAergic cells following FGF8 treatment.

*Figure 6 continued on next page*

*Figure 6 continued*

**Figure supplement 1.** Enrichment analysis of Gene Ontology (GO) terms in distinct NP or neuronal clusters.

**Figure supplement 1—source data 1.** Gene Ontology (GO) enrichment analysis in NP and neuronal clusters.

FGF8 on A/P patterning, we quantified the percentage of cells positive for selected A/P master genes (*Cadwell et al., 2019*; *Figure 7A–C*) in both glutamatergic NP (clusters 2/5/12/14/15) or neurons (clusters 1/3/4). Anterior genes, including *PAX6, ETV1, ZIC1*, and *ZIC3*, showed increased expression in progenitors and/or neurons following FGF8 treatment (*Figures 6B and 7B–D*). *Vice versa*, and consistent with the upregulation of anterior markers, posterior ones such as *NR2F1, FGFR3*, and *EMX1* were efficiently down-regulated by FGF8 in both NPs and neurons (*Figures 4A and 7B–D*). However, some anterior genes (e.g. *ETV5*) and posterior genes (e.g. *CRYM, TSHZ2, EMX2*, and *ODZ3*) displayed minimal changes (*Figure 7B and C*). These data suggest that FGF8 selectively modulates a subset of areal-specific genes within glutamatergic clusters, while other markers may be regulated *via* alternative mechanisms. To achieve a more comprehensive and unbiased assessment of A/P identity in telencephalic organoids, we used a SingleR approach to compare the transcriptomic profiles of control and FGF8-treated organoids against a reference dataset of distinct human fetal brain regions

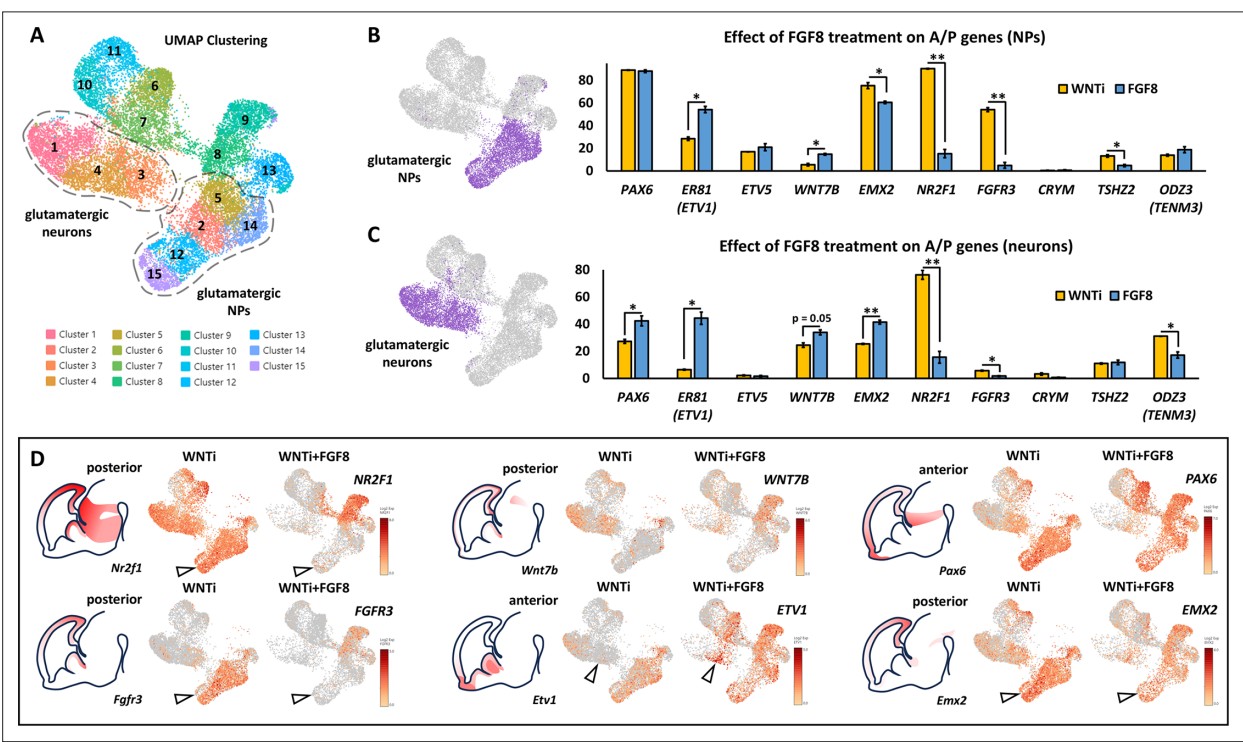

**Figure 7.** FGF8-dependent acquisition of distinct antero-posterior areal identities in human telencephalic organoids. (**A**) Glutamatergic neural progenitor (NPs; clusters 2/5/12/14/15) and glutamatergic neurons (clusters 1/3/4) are highlighted on the UMAP projection of day69 organoid scRNAseq data. (**B,C**) Images on the right display the clusters selected for analysis, as described in A. The graphs show the percentages of cells expressing anterior-posterior (A/P) cortical markers within the two highlighted cluster groups: NPs (top graph) and neurons (bottom graph). The percentage of cells positive for anterior markers (*PAX6, ER81, ETV5*) and posterior markers (*WNT7B, EMX2, NR2F1, FGFR3, CRYM, TSHZ2, ODZ3*) is shown in yellow for control (WNTi) organoids and in blue for FGF8-treated (WNTi + FGF8) organoids. (**D**) Expression level of key posterior (*NR2F1, FGFR3, WNT7B, EMX2*) and anterior (*ETV1, PAX6*) genes in UMAP projections of WNTi or WNTi + FGF8 day69 organoid samples, as indicated. Black arrowheads in the *NR2F1* and *FGFR3* UMAP projections point to decreased expression in proliferating glutamatergic progenitors upon FGF8 treatment, while arrowheads in the *ETV1* UMAP projection indicate increased expression in FGF8-treated glutamatergic neurons. Brain schematics with gene expression patterns are based on embryonic day 13.5 staining data from the Mouse Allen Brain Atlas.

The online version of this article includes the following source data and figure supplement(s) for figure 7:

**Source data 1.** Percentage of cells expressing antero-posterior cortical markers in WNTi and WNTi + FGF8 organoids.

**Figure supplement 1.** FGF8-dependent acquisition of different antero-posterior areal identities in human telencephalic organoids.

**Figure supplement 1—source data 1.** SingleR cluster annotation analysis of human organoid areal identity.

at 16 post-conception weeks (PCWs), including parietal, motor, prefrontal, somatosensory, temporal, and visual cortices (*Bhaduri et al., 2020*; *Eze et al., 2021*; *Nowakowski et al., 2017*; *Speir et al., 2021*; *Figure 7—figure supplement 1C*). Notably, WNTi and WNTi + FGF8 organoid samples (each subdivided into NP clusters or differentiated neuron clusters) showed high similarity to 16 PCW prefrontal and somatosensory cortices (*Figure 7—figure supplement 1C*, *left graph*). Specifically, by visualizing the SingleR annotation scores between organoid samples and fetal brain areas (*Figure 7—figure supplement 1C*, *right graph*), we found that neural progenitors, regardless of FGF8 treatment, predominantly resembled the fetal somatosensory cortex, while differentiated neurons exhibited greater similarity to either the prefrontal cortex or somatosensory cortex. Despite these findings, FGF8 treatment had a detectable effect on A/P identity, marked by a slight increase in the similarity of WNTi + FGF8 NPs to the transcriptional profile of the prefrontal cortex. Conversely, WNTi + FGF8 NPs and neurons showed reduced similarity to the fetal visual cortex and temporal areas (*Figure 7—figure supplement 1C*, *right graph*). Collectively, our data suggest that FGF8 modulates areal patterning by promoting the expression of genes associated with an anterior (prefrontal) cortical identity while reducing markers of a posterior (visual) identity.

## Discussion

### A hybrid 2D/3D cerebral organoid protocol enables rapid and reproducible generation of telencephalic progenitors and neurons

Over the last decade, 3D human organoids have emerged as a powerful tool for modeling both normal and pathological embryonic brain development. In this study, we adapted previous protocols to introduce a novel experimental approach that combines efficient 2D NP induction (*Chambers et al., 2009*) with 3D culture in spinning bioreactors (*Qian et al., 2018*). While dual SMAD inhibition in 2D allows rapid neural induction in just 7 days, subsequent steps in 3D culture promote the formation of reproducible brain-like cellular architectures, including rosettes and neural epithelia, which gradually mature into neurons organized in a cortical plate-like structure. As early NPs are dissociated and re-aggregated to form 3D embryoid bodies, this hybrid 2D/3D method could be particularly beneficial for experimental approaches that aim to mix different NP types or cells with distinct genetic backgrounds. It is important to note that the use of a ROCK inhibitor was necessary during early NP dissociation, as significant cell death occurred within 24 hr in its absence, indicating that early human progenitor cells are susceptible to apoptosis when cell contacts are lost, similar to undifferentiated hiPSCs. In late culture samples, our scRNAseq data revealed the presence of distinct types of human NPs, including *HOPX*+ outer radial glia (RG) cells (*Pollen et al., 2015*) and, notably, *CRYAB*+ apical truncated RG cells, which, to our knowledge, have only been identified in a few protocols (*Nowakowski et al., 2016*). This high cellular heterogeneity is likely promoted by the dual SMAD inhibition approach combined with WNT inhibition, a condition known to enhance NP amplification and diversity (*Rosebrock et al., 2022*).

### FGF8-induced cellular heterogeneity leads to the formation of segregated co-developing domains in multi-regional organoids

The regionalization of the embryonic brain is a multi-step process that operates both sequentially and simultaneously, primarily regulated by localized sources of various signaling molecules that function as organizing centers to pattern neighboring regions and create distinct molecular domains (*Borello and Pierani, 2010*; *Takahashi and Liu, 2006*). From E8.5 in the mouse, FGF8 is expressed at the boundary between the midbrain and hindbrain, playing a crucial role in posterior brain patterning and strongly promoting midbrain identity (*Harada et al., 2016*; *Liu et al., 1999*; *Martinez et al., 1999*). This is why protocols designed to generate midbrain neurons *in vitro* employ early FGF8 treatment (*Chambers et al., 2009*; *Perrier et al., 2004*). Consistently, our findings revealed that the early addition of FGF8 to the culture media resulted in the abolition of *FOXG1* expression, thereby preventing telencephalic induction. Importantly, our data indicate that in human brain organoids, day10-11 represents the optimal window for FGF8 application, allowing for the preservation of *FOXG1* expression and telencephalic identity while effectively modulating FGF8 target genes such as *NR2F1*. In contrast, initiating FGF8 treatment from day20 maintained *FOXG1* expression but was less effective in regulating *NR2F1*. Consequently, we only treated organoids with FGF8 after a minimum of 10 days of neural induction

to avoid early interference with telencephalic development. Despite this precaution, some regions resembling diencephalic and mesencephalic structures still formed in our organoids, likely due to the presence of unspecified NPs at day10, which might remain responsive to the mesencephalon-inducing effects of FGF8. We propose that FGF8 can play multiple roles in human cells *in vitro*, depending on the competence, developmental state, and regional identity of NPs exposed to it. Early in development, when acting on unspecified progenitors, FGF8 serves as a potent inducer of posteriorized identity, capable of specifying discrete diencephalic/mesencephalic-like domains in brain organoids. However, as development progresses and *FOXG1* expression becomes established, consolidating a telencephalic fate, FGF8 acts as a regulator of A/P and D/V identities in forebrain cells. Interestingly, the ability of FGF8 to modulate target genes can also vary based on the regional identity of the FGF8-exposed cells.

The co-presence of various regional domains in FGF8-treated organoids - specifically dorsal and ventral telencephalon, as well as diencephalic and mesencephalic regions - represents a significant outcome. One of the major challenges in the field of *in vitro* models of human brain development is successfully reproducing the interactions between distinct brain regions. Unguided organoids tend to spontaneously develop multiple brain regions (*Lancaster et al., 2013*; *Lancaster and Knoblich, 2014*; *Renner et al., 2017*), but this occurs in a stochastic manner, resulting in weak reproducibility across different batches (*Velasco et al., 2019*). A more reliable strategy involves utilizing signaling molecules, such as morphogens and chemical drugs in the culture medium, to control the patterning of organoids into specific structures, like the cortex, ventral telencephalon, hypothalamus, and thalamus. However, organoids specified to distinct identities often require further manipulation, such as the fusion of different structures in multiwell plates and/or embedding them together in Matrigel to create 'assembloids' (*Paşca, 2019*). This additional step can be time-consuming and may introduce further variability, even though it is essential for studying neuronal migration and facilitating the formation of inter-regional neuronal connections.

In this study, we demonstrate that the addition of patterning cues (FGF8 in this case) to the culture medium - even when applied in a non-polarized manner - is sufficient to instruct the development of additional regional fates that co-exist and co-develop within the same organoid while maintaining spatial segregation. Notably, we identified FOXG1+ telencephalic areas containing both TBR1+ dorsal cortical neurons and ventral GABAergic cells, alongside more posterior OTX2+ FOXG1- diencephalic/mesencephalic-like domains. By enhancing organoid complexity, FGF8 signaling enables the formation of more biologically realistic models of human brain assembly *in vitro*. Multi-regional organoids present an opportunity to investigate how different brain regions self-organize and interact, thus eliminating the need for manual assembly of pre-patterned organoids into assembloids. We propose that the simple addition of instructing cues (morphogens) to the culture medium can enhance the complexity of brain organoids, providing a compromise between maintaining a certain degree of spontaneous self-organization while inducing multiple brain regions in a reproducible manner.

## FGF8 signaling as a key driver of lateral ganglionic eminence specification in D/V telencephalic patterning

Within the telencephalon, FGF8 forms a gradient from its anterior source, the ANR, acting as a morphogen that triggers distinct cellular responses relative to its concentration. In mice, reduced Fgf8 levels cause progressive telencephalic hypoplasia (*Storm et al., 2006*), as described in *Fgf8* hypomorphic and conditional mutants, which show smaller telencephalons due to decreased proliferation, increased apoptosis, and altered expression of areal patterning genes such as *Nr2f1*, *Pax6*, *Emx2*, and *Sp8* (*Storm et al., 2006*; *Storm et al., 2003*; *Garel et al., 2003*). Notably, severe *Fgf8null* mutants exhibit a marked reduction of antero-ventral structures, likely impacting adjacent signaling centers expressing *Bmp4*, *Wnt8b*, and/or *Shh*. Conversely, conditional *Fgf8* mutants, where *Fgf8* is inactivated at a later stage, display a milder phenotype, with reduced frontal cortex and ventral structures and an expanded dorso-posterior molecular profile (*Storm et al., 2006*; *Garel et al., 2003*). Severe phenotypic alterations and cross-regulation between anterior (FGF), dorsal (BMP, WNT), and/or ventral (SHH) patterning centers in genetic loss-of-function (LOF) animal models obscure the specific role of FGF8 signaling in telencephalic development. Therefore, a system with controlled FGF8 signaling modulation is preferable.

In this study, we investigated the exclusive, long-term effects of FGF8 signaling on human cerebral organoids by enabling proper differentiation of FOXG1+ telencephalic cells and assessing molecular and cellular changes across developmental stages. Transcriptomic analysis of FGF8-treated *versus* untreated cerebral organoids revealed a clear FGF8-mediated induction of ventral telencephalic genes, suggesting that one of the primary roles of FGF8 signaling within telencephalic territories may be the promotion of ventral identity. Consistently, we observed upregulation of ventral genes such as *ASCL1*, *DLX2*, and *PBX3* and simultaneous downregulation of dorsal genes like *EMX1*, *NEUROG2*, *SOX5*, and *LHX2*. As a result, FGF8-treated organoids showed an altered balance between glutamatergic neurons (expressing *NEUROD6*, *NEUROD2*, *CTIP2*, *TBR1*, *SATB2*, *NF1A*, and *SLC17A7*) and GABAergic neurons (expressing *EBF1*, *GSX2*, *PBX3*, and *GAD1*). Supporting these findings, functional assays using multi-electrode arrays (MEA) indicated reduced spontaneous network activity and decreased electrical signal propagation in FGF8-treated organoids compared to controls. This electrophysiological profile, with lower firing and network burst rates, aligns with a reduction in excitatory glutamatergic neurons and an increase in inhibitory GABAergic neurons (*Mossink et al., 2022*) and supports an FGF8-induced shift toward ventral GABAergic identity at the expense of a dorsal glutamatergic identity in telencephalic organoids. Acute FGF8 modulation in mice, achieved by implanting FGF8-soaked beads into E9.5 telencephalic explants, was shown to induce the expression of ventral markers like *Ascl1* and *Dlx2* while repressing the dorsal cortical marker *Emx1* (*Kuschel et al., 2003*). Our data in human organoids align with these mouse findings and suggest that the absence of FGF8 may hinder the induction of ventral markers *in vitro* as well as the formation of ganglionic eminence structures *in vivo* (*Storm et al., 2006*).

Interestingly, we observed that certain genes upregulated by FGF8, such as *MESI2*, *TLE4*, and *PBX3*, are specific to the ventral lateral ganglionic eminence (LGE), a region ventral to the lateral pallium, and, particularly, are characteristic of striatal precursors (*Shi et al., 2021*). This FGF8-induced effect does not extend to medial ganglionic eminence (MGE) markers such as *SHH*, *LHX8*, and *NKX2-1*. To our knowledge, the striatal-promoting effect of FGF8 has been little documented before; it could be interesting to test the combined influence of FGF8 with SHH, known to promote medium spiny neuron production *in vitro* (*Delli Carri et al., 2013*). Notably, the 'ventral' marker ASCL1, which is upregulated by FGF8 treatment in our organoids, is also expressed in a subset of glutamatergic progenitors in rodents (*Britz et al., 2006*) and humans (*Alzu'bi and Clowry, 2019*). This suggests that ASCL1 is not exclusively a marker of cells of ventral origin, particularly within the human telencephalon (*Delgado et al., 2022*; *Kim et al., 2023*), where lineage-tracing studies reveal that dorsal progenitors expressing LGE-like markers can also differentiate into GABAergic neurons (*Delgado et al., 2022*; *Alzu'bi et al., 2017b*). In this context, FGF8 may enhance an inherent GABAergic-producing capacity in human telencephalic progenitors by promoting the expression of LGE markers. Thus, FGF8-induced ventral LGE identity in human organoids may reflect a species-specific feature: the intrinsic potential of human glutamatergic progenitors to produce GABAergic neurons with an LGE-like molecular profile.

## FGF8-mediated control of A/P areal identity and unified role of FGF8 as an 'antero-ventral' inducer

FGF8 is a well-established key regulator of anterior *versus* posterior identity, directing cortical area specification in the mouse neocortex (*Alfano and Studer, 2013*). Its gradient diffuses from anterior to posterior along the neocortical epithelium, promoting prefrontal while inhibiting more occipital cortical areas, like the visual cortex. Although human cerebral organoids do not develop segregated functional areas as in the developing neocortex (*Cadwell et al., 2019*; *Bhaduri et al., 2020*), we observed that FGF8 can downregulate the expression of posterior cortical genes, like *FGFR3* and *NR2F1*, while inducing anterior ones, such as *ETV1*. This suggests that beyond controlling D/V cell identity, FGF8 may also influence A/P areal-specific gene expression in human cerebral organoids. However, the global identity of both FGF8-treated and untreated organoids, based on transcriptional similarity to human fetal brain areas, remained aligned with somatosensory and pre-frontal cortical regions, indicating only a limited effect on full A/P areal specification. Optimizing FGF8 doses or timing could enhance A/P areal identity control in telencephalic organoids. A promising approach may involve introducing a polarized FGF8 source to better mimic physiological gradients, as done with SHH-expressing organoids (*Cederquist et al., 2019*), where an endogenous signaling center

generates distinct D/V identities in a dose-dependent manner. Establishing a localized FGF8 source adjacent to polarized forebrain organoids could expose telencephalic cells to varying concentrations of FGF8, potentially improving *in vitro* modeling of areal patterning by inducing distinct A/P areal identities and ultimately establishing neocortical axes. The introduction of discrete morphogen sources in polarized organoids has been explored in recent reports (*Bosone et al., 2024*; *Xue et al., 2024*).

In summary, our study shows that FGF8 influences both D/V and A/P regional identity in telencephalic organoids. Although we analyzed the A/P and D/V axes separately, these induction and patterning processes are likely interconnected. For instance, specific neocortical areas along the A/P axis have varying populations of ventrally-generated GABAergic interneurons (*Molnár et al., 2019*). Interestingly, the anterior-most prefrontal cortex has a higher relative proportion of GABAergic neurons (*Molnár et al., 2019*; *Zhong et al., 2018*), although the mechanisms originating this property remain unclear. Given the antero-ventral position of the ANR in the early telencephalon, FGF8 could be considered an 'antero-ventral' inducer. This might explain the dual role of FGF8 in cerebral organoids, promoting both ventral (LGE-like) and anterior brain identities, generating new hypotheses about the role of FGF8 in linking cortical areal identity with the abundance and subtype of GABAergic interneurons. It is tempting to speculate that, rather than relying solely on migration from ventral regions, these interneurons could be locally generated by cortical progenitors exposed to distinct, areal-specific doses of FGF8.

## FGF8 signaling impacts NDD-related developmental trajectories

Through long-term treatment of human organoids, we identified a correlation between FGF signaling activation and the modulation of several developmental and/or neurodevelopmental disorder (NDD)-related genes. Among the FGF8-regulated genes, *NR2F1* was the most significantly affected, suggesting that it may be a primary effector of FGF8 signaling in telencephalic development. In addition to *NR2F1*, other FGF8-responsive genes detected in the DEG analysis are implicated in human brain malformations and/or NDDs. For example, dysregulation of *FGFR3* leads to Thanatophoric dysplasia, a fatal form of chondrodysplastic dwarfism, characterized by temporal lobe enlargement, abnormal sulci, and hippocampal dysplasia, resulting in cognitive impairments and reduced synaptic plasticity in both patients and mouse models (*Hevner, 2005*). *ZIC1*, which was significantly up-regulated in FGF8-treated organoids, is implicated in complex syndromes involving cortical, callosal, and cerebellar malformations associated with intellectual disability (*Twigg et al., 2015*; *Vandervore*

**Table 1.** Primer sequences used.

| gene | forward sequence | reverse sequence |
| --- | --- | --- |
| *DUSP6* | AACCTGTCCCAGTTTTTCCCT | GCCAAGCAATGTACCAAGACAC |
| *EN2* | TCTACTGTACGCGCTACTCG | CGCTTGTCCTCTTTGTTCGG |
| *ETV1* | CCAATAGTCAGCGTGGGAGAA | TTCTGCAAGCCATGTTTCCTGT |
| *ETV4* | TCAAACAGGAACAGACGGACTT | AGGTTTCTCATAGCCATAGCCCA |
| *ETV5* | ACACGGGTTCCAGTCACCAA | GCTGCTGGAGAAATAACCCCC |
| *FOXG1* | GCCAAGTTTTACGACGGGAC | AAGGGGTTGGAAGAAGACCCC |
| *GAPDH* | CGTGGAAGGACTCATGACCA | CAGTCTTCTGGGTGGCAGTGA |
| *NANOG* | CAAAGGCAAACAACCCACTT | TCTGCTGGAGGCTGAGGTAT |
| *NR2F1* | TGGCAATGGTAGTTAGCAGCT | TTGAGGCACTTCTTGAGGCG |
| *OCT4* | GTGGAGGAAGCTGACAACAA | ATTCTCCAGGTTGCCTCTCA |
| *OTX2* | CCTCACTCGCCACATCTACT | CTTGGTGGGTGGGTTTGGAG |
| *PAX6* | AGTGAATCAGCTCGGTGGTGTCTT | TGCAGAATTCGGGAAATGTCGCAC |
| *SIX3* | CAGCAAGAAACGCGAACTGG | AATGGCCTGGTGCTGGA |
| *SOX2* | AGGATAAGTACACGCTGCCC | TAACTGTCCATGCGCTGGTT |
| *SPRY4* | CGGCTTCAGGATTTACACAGAC | CTGCAAACCGCTCAATACAGG |
| *β-Actin* | CTTCGCGGGCGACGAT | ACATAGGAATCCTCCTGACCC |

et al., 2018). Moreover, haploinsufficiency of *NFIB* results in macrocephaly and impaired intellectual development, similar to what has been described in *Nfib* mutant mice (**Schanze et al., 2018**). We also identified other disease genes, including *AUTS2* (linked to autism; **Hori et al., 2021**), *FOXP2* (associated with speech and language disorders; **Fisher and Scharff, 2009**), and *SOX5* (linked to developmental delay or intellectual disability; **Zawerton et al., 2020**). Finally, *FOXG1*, normally induced by FGF8, is implicated in a range of brain disorders, including the congenital variant of Rett syndrome, infantile spasms, microcephaly, autism spectrum disorder, and schizophrenia (**Hou et al., 2020**). In our organoid model, however, early FGF8 treatment led to a decrease in *FOXG1* levels rather than an increase, suggesting that FGF8-mediated regulation of *FOXG1* in NPs and neurons could be time- and region-dependent. In summary, our findings highlight an FGF8-dependent effect on regional identity, along with the modulation of several NDD-related targets, including an evolutionarily conserved *FGF8-NR2F1* molecular axis.

In conclusion, we propose that FGF8-mediated modulation of key developmental genes guides the developmental trajectories of human brain cells along the D/V and A/P telencephalic axes. Diverse genes and signaling pathways may converge to orchestrate shared cellular and molecular processes, resulting in similar or overlapping phenotypes in neurodevelopmental pathologies (**Parenti et al., 2020**). We speculate that a disruption in an FGF8-dependent molecular framework could lead to NDD phenotypes by altering the expression of NDD-related genes and/or interfering with the fundamental processes of A/P and D/V neuronal specification during early brain development.

# Materials and methods

## Key resources table

| Reagent type (species) or resource | Designation | Source or reference | Identifiers | Additional information |
|---|---|---|---|---|
| Cell line (*Homo sapiens*) | HMGU1; Human induced pluripotent stem cells | kind gift of Dr. Drukker | HMGU1; source cells: BJ (ATCC CRL-2522) | MTA approval was obtained from the Helmholtz Zentrum München (HMGU), Germany |
| Peptide, recombinant protein | Recombinant Human/Mouse FGF-8b Protein | R&D | 423-F8-025/CF | (100 ng/ML) |
| Peptide, recombinant protein | BDNF | PeproTech | #450–02 | 10 ng/mL |
| Commercial assay or kit | hPSC genetic Analysis kit | Stem Cell Technologies | #07550 | hiPSCs culture |
| Commercial assay or kit | NucleoSpin RNA II columns | Macherey-Nagel | 740902.50 | hiPSCs culture |
| Commercial assay or kit | Reverse Transcriptase Core Kit | Eurogentec | RT-RTCK-03 | hiPSCs culture |
| Chemical compound, drug | Matrigel | Corning | 354234 | 5–10 µl matrigel dissolved in 1 ml cold DMEM-F12 for each well of a 6-well culture plate |
| Chemical compound, drug | mTeSR1 medium | STEMCELL Technologies | #85850 | hiPSCs culture |
| Chemical compound, drug | ROCK inhibitor Y-27632 | MedChemExpress; or Stem Cell Technologies | MedChemExpress HY-10583; or Stem Cell Technologies #72304 | 10 µM |
| Chemical compound, drug | DMEM/F12 | Thermo Fisher Scientific | #31331028 | hiPSCs culture |
| Chemical compound, drug | N-2 Supplement 100X | Thermo Fisher Scientific | 17502–048 | hiPSCs culture |
| Chemical compound, drug | B-27 Supplement 50X, minus vitamin A | Thermo Fisher Scientific | 12587–010 | hiPSCs culture |
| Chemical compound, drug | LDN-193189 | Sigma | SML0559-5MG | 0.25 µM |
| Chemical compound, drug | SB-431542 | Sigma | S4317-5MG | 5 µM |
| Chemical compound, drug | XAV-939 | Stem cell technologies | 72674 | 2 µM |
| Antibody | CTIP2, FOXG1, GAD1, OTX2, MAP2, ASCL1, NR2F1, NTUB (ACETILATED TUBULIN), OCT-3/4, PAX6, SATB2, SOX2, TBR1, TUJ1, SLC17A6, SLC17A7 antibodies | See *Table 2* for supplier information and antibody host species | See *Table 2* for reference codes | See *Table 2* for antibody dilution |
| Sequence-based reagent | DUSP6, EN2, ETV1, ETV4, ETV5, FOXG1, GAPDH, NANOG, NR2F1, OCT4, OTX2, PAX6, SIX3, SOX2, SPRY4, B-ACTIN primers | This paper | PCR primers | See *Table 1* for sequence of forward and reverse primers |

Human induced pluripotent stem cells (hiPSCs) used in this study are the HMGU1 cell line, derived from fibroblasts established from skin taken from normal foreskin from a neonatal male (ATCC number CRL-2522, designation BJ), mycoplasma-free and a kind gift of Dr. Drukker. MTA approval was

obtained from the Helmholtz Zentrum München (HMGU), Germany. HMGU1 hiPSCs were cultured on Matrigel-coated plates (Corning, 354234; 5–10 µl matrigel dissolved in 1 ml cold DMEM-F12 for each well of a six-well culture plate) in mTeSR1 medium (STEMCELL Technologies; #85850). Medium was changed daily. HMGU1 cells were passaged with Versene (Thermo Fisher Scientific; 15040066) as previously described *Beers et al., 2012*; briefly, cells were washed with 1 ml PBS1x (Thermo Fisher Scientific 14190169; or Sigma D8537), treated with Versene for up to 5 min, then detached by gently pipetting with 1 ml mTeSR1 medium. Alternatively, when single cell dissociation and cell counting were needed, HMGU1 cells were passaged with Accutase (Sigma; A6964). Single cells passaging required addition of 10 µM dihydrochloride ROCK inhibitor Y-27632 (MedChemExpress HY-10583; or Stem Cell Technologies #72304) to prevent apoptosis. ROCK inhibitor was removed 24 hr after dissociation. For immunostaining, cells were grown on glass coverslips (Epredia X1000 Round Coverslip dia. 13 mm, Product Code: 10513234) with the same medium and coating treatment.

## hiPSCs quality controls

HMGU1 cells were checked for mycoplasma contamination once per year, while amplifying and preparing multiple cryovial aliquots that were used during the following months. Genetic/chromosomic analysis was performed by using the hPSC genetic Analysis kit (Stem Cell Technologies; #07550) by following the manufacturer's instructions. Two different hiPSC lines (WT-T12, kind gift of Dr. Magdalena Laugsch, and PGP1, purchased from Synthego) were used as controls to be compared with HMGU1 cells. Briefly, hiPSC cell pellets were lysed with lysis buffer (1 M TRIS HCL pH8, 5 M NaCL, 0.5 M EDTA, 10% SDS in milli-Q $H_2O$) supplemented with 10 mg/mL Proteinase K (Sigma, 0311580); lysis incubation was performed at 58 °C for 30 min. Genomic DNA was obtained by Isopropanol precipitation and 70% Ethanol cleaning. We detected frequent mutations in the HMGU1 cell line, and found that extended culture of hiPSCs caused chromosomal abnormalities to further accumulate in sensitive regions. To prevent this, we did not use HMGU1 cells after passage 26. Finally, pluripotency was checked by OCT4 immunostaining.

## Neural differentiation into telencephalic organoids

HMGU1 cells were seeded in Matrigel-coated 24 well-plates at a density of 25,000–50,000 cells per well and cultured in mTeSR1 medium till high confluency, which is key for efficient neural induction in this protocol (*Chambers et al., 2009*). When highly confluent (90–95% surface covered), hiPSCs were washed with pre-warmed DMEM/F12 (Thermo Fisher Scientific; #31331028) then medium was switched to neural progenitor patterning medium (NPPM). NPPM consisted of DMEM/F12 supplemented with N2 (N-2 Supplement 100X; Thermo Fisher Scientific 17502–048), B27 (B-27 Supplement 50X, minus vitamin A; Thermo Fisher Scientific 12587–010), GlutaMAX (Thermo Fisher Scientific 35050038), Non-Essential Amino-Acid (NEAA; Thermo Fisher Scientific 11140–035), Sodium Pyruvate (NaPyr; Thermo Fisher Scientific #11360070), 50 µM 2-mercaptoethanol (Thermo Fisher Scientific 31350–010), 2 µg/mL Heparin (Sigma H3149) and Penicillin/Streptomycin (Biozol ECL-EC3001D). The latter could be substituted by Antibiotic/Antimycotic solution (Sigma A5955). Neural induction was boosted by adding a BMP inhibitor (0.25 µM LDN-193189; Sigma SML0559-5MG) and a TGFβ inhibitor (5 µM SB-431542; Sigma S4317-5MG) for the control condition ('CTRL' in figures); telencephalic induction also required use of a WNT inhibitor (2 µM XAV-939; Stem cell technologies 72674; 'WNTi' in figures). On day 7 (max on day 8) of the protocol, cells were dissociated by Accutase (Sigma; A6964) and seeded in 96 U-bottom well plates (Corning #7007; or Sarstedt 83.3925500) at a concentration of 60,000 cells/well, in NPPM medium containing ROCK inhibitor. A gentle centrifugation of the 96-well plate (1 min at 1000 rpm) allowed quick accumulation of the cells at the bottom of the wells. The day after, the embryoid bodies (EBs) were collected and transferred in Matrigel droplets (Corning 354234; 70 µl Matrigel for 16 EBs) to generate 'cookies', distributed on parafilm and incubated 30 min at 37 °C for Matrigel to solidify, then transferred in SpinΩ mini-rotors (*Qian et al., 2018*) in NPPM medium (still supplemented with SB + LDN ± XAV; 2.5 ml per well in a 12-well plate). On day9 (max on day10), most of the medium was removed and replaced with NPPM without SB-431542 and LDN-193189, but still containing WNT-inhibitor. Starting on day10, FGF8 (100 ng/ML; Recombinant Human/Mouse FGF-8b Protein; R&D 423-F8-025/CF) could be added to NPPM medium, that was then changed every 2–3 days, to create the treated condition (referred to in the text and figures as 'WNTi + FGF8'). If medium

showed excessively yellow color indicating pH acidification, cookies were diluted by redistributing in multiple wells to have a maximum of five cookies per well. At day 20, medium was switched for Neural Differentiation Medium (NDM) containing 50% DMEM-F-12 medium and 50% Neurobasal medium (ThermoFisher scientific 21103049) supplemented with N2, Glutamax, NEAA, NaPyr, 50 μM Beta-mercaptoethanol, 2 μg/mL Heparin and Insulin (25 μl per 100 ML medium; Sigma #19278). At day30 (max at day35), NDM was substituted with Long-term pro-Survival Medium (LTSM) consisting in Neurobasal medium supplemented with N2, B27 without Vitamin A, Glutamax, NEAA, NaPyr, 50 μM Beta-mercaptoethanol, 2 μg/mL Heparin, 1% Serum (Fetal Bovine Serum, ThermoFisher scientific 10270–106; inactivated for 30 min at 56 °C then aliquoted and stored at –20 °C), 10 ng/mL BDNF (PeproTech #450–02) and dissolved Matrigel (final concentration in the medium: 0.1%). Medium was changed every 3–4 days, until the desired differentiation stage. Starting at day50-60, bigger organoids were sliced with a sterile blade under the stereomicroscope (typically once per month), which allows for better survival and reduces the formation of an inner necrotic core (*Qian et al., 2020*). Organoids were cultured up to 110 days.

## Real time qRT-PCR

Cell pellets or organoids (typically cell pellets from two independent wells or three organoids) were pooled and spun in 1.5 ml Eppendorf tubes, then frozen at –80 °C. Total RNA was extracted with NucleoSpin RNA II columns (Macherey-Nagel; 740902.50). RNA quantity and quality were assessed with Nanodrop and gel electrophoresis. For each sample, 200/500 ng of total RNA were reverse transcribed using random nonamers (Reverse Transcriptase Core Kit, Eurogentec, qRT-RTCK-03); qRT-PCR was performed using GoTaq SYBR Green qPCR Mix (Promega A6001) or KAPA SYBR$^R$ FAST (KAPA Biosystems; KK4610) on LightCycler (Roche). cDNA (stored at –20 °C) was diluted so that each reaction contained 2 ng. Amplification take-off values were evaluated using the built-in LightCycler relative quantification analysis function, and relative expression was calculated with the $2^{-\Delta\Delta Ct}$ method as previously described (*Bertacchi et al., 2015a*), normalizing to the housekeeping genes *GAPDH* or *β-Actin*. Standard errors for error bars were obtained from the error propagation formula (*Bertacchi et al., 2015a*). For each genotype/time point, 2–3 organoids/cell pellets (biological replicates) were pooled together before RNA extraction, while at least two reactions were assembled per sample/gene analyzed during qRT-PCR amplification (technical replicates).

**Table 2.** Primary antibodies used.

| Primary antibody | species | Concentration used | Brand | ref |
|---|---|---|---|---|
| CTIP2 | Rat monoclonal | (1:1000) | Abcam | ab18465 |
| FOXG1 | Rabbit polyclonal | (1:1000) | Abcam | ab18259 |
| GAD1 | Mouse monoclonal | (1:1000) | Millipore | MAB5406 |
| OTX2 | Goat polyclonal | (1:1000) | R&D | AF1979 |
| MAP2 | Mouse monoclonal | (1:1000) | Sigma | M4403 |
| ASCL1 (MASH1) | Mouse monoclonal | (1:1000) | BD Biosciences | 556604 |
| NR2F1 (COUP TF1) | Rabbit monoclonal | (1:1000) | Abcam | ab181137 |
| NTUB | Mouse monoclonal | (1:1000) | Sigma | T6793 |
| OCT-3/4 | Mouse monoclonal | (1:500) | Santa Cruz | sc-5279 |
| PAX6 | Rabbit polyclonal | (1:1000) | Millipore | AB2237 |
| SATB2 | Mouse monoclonal | (1:1000) | Abcam | ab51502 |
| SOX2 | Mouse monoclonal | (1:1000) | R&D | MAB2018 |
| TBR1 | Rabbit polyclonal | (1:1000) | Abcam | ab31940 |
| TUJ1 (TUBB3) | Rabbit polyclonal | (1:1000) | BioLegend | 802001 |
| SLC17A6 (VGLUT2) | Guinea Pig polyclonal | (1:1000) | Millipore | AB2251 |
| SLC17A7 | Rabbit polyclonal | (1:1000) | Invitrogen | 48–2400 |

## Immunostaining on cryostat sections

Organoids were collected in 2 ml Eppendorf tubes and fixed in 2 ml of 4% PFA at 4 °C for 3–4 hr in gentle agitation, then washed twice with PBS 1 X and dehydrated in 10% sucrose (Sigma-Aldrich, S9378-1KG; dissolved in PBS1x; minimum 4 hr to maximum over-night) followed by 25% sucrose (dissolved in PBS1x; overnight at 4 °C). Most of the sucrose was removed and substituted with OCT resin (Leica Tissue Freezing Medium, 14020108926; or Cryomatrix, 6769006) with a 10 min wash with gentle rotation, then organoids were transferred in embedding molds in clean OCT and stored at –80 °C. Cryostat 12 µm sections (Leica cryostat, model: CM3050S) were collected on glass slides (Thermo Fisher Scientific, Superfrost Plus, J1800AMNZ, or VWR SuperFrost Plus, 631–0108) and stored at –80 °C. Prior starting immunostaining, sections were dried at RT for 10–15 min, then washed two times with PBS1x (10 min each) to remove traces of OCT resin. All antibodies required antigen retrieval prior to incubation (unmasking: 10 min at 95 °C in pH = 6 0.1 M sodium citrate solution). After a 5-min PBS1x wash to remove unmasking solution, pre-blocking solution was added on the slides for 1 hr, containing PBS1x with 5% serum (Sheep serum, Sigma, S2263-100ML; in case of primary antibodies raised in sheep or goat: Newborn Calf Serum, Thermo Fisher Scientific, 16010–167) and 0.3% Triton (Sigma-Aldrich, T8787-250ML). For primary antibody incubation (from a minimum of 4 hr at RT to a maximum of over-night at 4 °C), blocking solution consisted in PBS1x supplemented with 1% serum and 0.1% Triton. Primary antibodies used in this study are listed in *Table 2*. Alexa Fluor 488, 555, 594, and 647 anti-mouse, anti-rabbit, anti-rat, anti-goat, anti-guinea pig or anti-sheep IgG conjugates (Thermo Fisher Scientific, all diluted at 1:500) were used as secondary antibodies (incubation time: from a minimum of 2 hr at RT to a maximum of over-night at 4 °C). Secondary antibody solution also contained 1:1.000 Hoechst 33342 for nuclei staining (Invitrogen, H3570). After final washes (3 times, 10 min each, in PBS 1 x), organoid sections were covered with mounting solution (80% glycerol, 2% N-propyl gallate in PBS1x) and glass coverslips, which were sealed with nail polish on the edges. Stained sections were stored at –20 °C. Images were acquired at an Apotome Zeiss, using the Axio-Vision software, and exported as TIF files. Alternatively, images were collected using a Vectra Polaris slide scanner (Akoya Biosciences) and then underwent artificial intelligence-driven analysis by HALO software (Indica Labs).

## Immunostaining of cultured cells

For immunostaining, cells were cultured on Matrigel coated round glass coverslip. The immunostaining protocol is the same used for cryostat sections, except fixation with 4% paraformaldehyde lasted only 15 min at room temperature, no antigen retrieval was performed, and PBS1x for washes always contained 1% serum to limit cell detachment.

## Statistical analysis

All data were statistically analyzed and graphically represented using Microsoft Office Excel software or GraphPad Prism (Version 7.00). Quantitative data are shown as the mean ± SEM. For cell percentage/number quantification after immunostaining, measurements were performed on at least five sections coming from two to three different organoids, unless otherwise stated. Organoid sections with damaged histology were excluded from any further analysis/processing; the inner necrotic core of the organoid, when present, was excluded from counting. Microscope images were processed with Photoshop or ImageJ software, by randomly overlapping fixed-width (100 µm) square boxes on the area of interest [*e.g.* organoid surface], then quantifying positive cells inside the boxes. When calculating percentages over the total cell number, the latter was quantified by counting Hoechst+ nuclei, unless otherwise specified. Data were analyzed using the Mann–Whitney U-test or two-tailed Student's t-test (when comparing two data groups), or by two-way ANOVA for comparison of three or more groups. Statistical significance was set as follows: *$p < 0.05$; **$p < 0.01$; ***$p < 0.001$.

## MEA recordings

For MEA recordings, we used the high-density MEA system (HD-MEA) from MaxWell (Model: MaxOne; Chips: MaxOne single-well chips). Prior recordings, organoids were plated on electrodes by following the MaxWell 'Brain organoid plating protocol, version 1', with few modifications. Briefly, MEA chips were cleaned with 1% Terg-a-zyme solution for minimum 2 hr at RT, washed in distilled water and sterilized with 70% Ethanol. A first coating with Poly-L-ornithine hydrobromide (20 µg/

ml in distilled water for 5 hr at 37 °C, Sigma P3655) was followed by a second one with Laminin (25 µg/ml in PBS1x, Santa Cruz, sc-29012). After a pre-incubation with LTSM medium (minimum 2 hr, maximum O/N at 37 °C; pre-conditioning medium was prepared as detailed in *Neural differentiation into telencephalic organoids* and supplemented with additional 5 µl Matrigel per 1 ml of medium) to pre-condition the chip surface, organoids were transferred on MEA chips (n=3/5 organoids per chip) and let deposit by gravity in the incubator, while avoiding any vibration for the next 24 hr. Medium was delicately changed every 3 days during a 2 week incubation on chip; during the second week, medium was gradually switched to LTSM that was prepared using BrainPhys (Stem Cell Technologies, #05790) as a base medium where to dilute the supplements. The organoids underwent complete adhesion within 1–2 weeks and interconnected with bundles of axons, a condition known to promote complex and oscillatory activity (*Osaki et al., 2024*). Medium was always changed 1 day before the recording to promote electric activity. Recordings were performed with the MaxOne software, by using in-built protocol 'Activity Scan Assay' to detect active areas and to measure mean firing rate and mean spike amplitude, followed by the 'Network Assay' to evaluate firing synchronicity and by the 'Axon Tracking Assay' to detect signal propagation along axonal tracts. Average data were obtained by recording from the whole chip surface, hence pooling active electrodes from 3/4 organoids per condition. For statistical analysis (GraphPad; 2-way ANOVA), recording data were pooled from control (WNTi) and treated (WNTi + FGF8) samples from three distinct batches (each including 2–4 organoids on the MEA chip active surface), including three different developmental stages *in vitro* (Batch1: days 131–143; Batch2: days 202–215; Batch3: days 153–187). For the detection of synchronous events (bursts) in the 'Network Assay', the threshold was automatically determined by the MEA system. This threshold depended on a fixed multiplying factor that was consistent across both control and treated samples; the factor was then multiplied by the basal average activity of each individual sample. This method allowed for the detection of bursts as synchronized activity emerging from the basal noise, which varies in every sample. For the 'Axon Tracking Assay', a total of 170 WNTi and 165 WNTi + FGF8 axonal tracts were detected and analyzed from the three batches (see *Figure 5*). An 'Axon Tracking Assay' analysis was also performed separately at different developmental stages to investigate axonal maturation (see *Figure 5—figure supplement 1*). For Bicuculline treatment on WNTi + FGF8 organoids (10 µM Bicuculline; Abcam, ab120107), 200 µl of medium were collected from the electrode, mixed with Bicuculline, added back to the MEA chip and gently mixed by pipetting. Recordings were conducted before (control condition) and after (treatment condition) the addition of Bicuculline, with a minimum drug incubation period of 15 minutes before starting the treatment condition recording.

## Cell dissociation for single-cell RNA sequencing and flow cytometry analysis

Day69 control (WNTi) and treated (WNTi + FGF8) organoids were dissociated using an enzymatic Papain Dissociation System (Serlabo technologies LK003150 or Worthington, #LK003150), by following manufacturer's instructions. Low-bind Eppendorf tubes were used to limit material loss. Two to three organoids per batch were isolated in Eppendorf tubes containing 1 ml of pre-warmed Papain solution supplemented with DNase. Total incubation lasted 70 min, with manipulations every ten minutes that consisted in gentle mixing by tube inversion (first 2 rounds) then gentle pipetting (following rounds), ending up with addition of Ovomucoid solution for Papain inactivation. Samples were filtered (Cell Strainer 40 µm Nylon, FALCON, 352340) to remove undissociated cells, spinned down (1200 rpm, corresponding to 170 *g*, for 5 minutes) and resuspended in PBS1x supplemented with 0.1% BSA (Jackson, 001-000-161). Cell viability upon dissociation was checked by Propidium Iodide (PI) staining in flow cytometry in an independent experiment, where Papain was compared with Accutase and with ReLeSR (Stem Cell Technologies, #100–0483). For flow cytometry, cells were stained with Propidium Iodide (40 µg/ML, Sigma, P4170) in PBS1x for 15 min at RT, washed twice with PBS1x supplemented with 1% Serum, then analysed with BD LSRFortessa and FACSDiva software (Becton Dickinson) to measure dying cells which incorporated PI. Cells were analysed on the basis of 10,000 total events (debris excluded). Cell viability was tested again by Trypan blue staining just prior cell counting for library preparation, and samples were considered suitable for single cell RNA sequencing only when damaged cells detected by Trypan blue or by PI staining were ≤10% of the total population.

## Single-cell RNA sequencing: library preparation and sequencing

Single-cell RNA sequencing experiment was performed by using 10 X Genomics technology. Dissociated cells were processed following the manufactures' instruction of Chromium Next GEM Single Cell 3′ Reagent Kits v3.1 (Dual Index). Briefly, they were resuspended in ice-cold PBS containing 0.1% BSA at a concentration of 1000 cells/µl, and approximately 17,400 cells per channel (corresponding to an estimated recovery of 10,000 cells per channel) were loaded onto a Chromium Single Cell 3′ Chip (10 x Genomics, PN 2000177) and processed through the Chromium controller to generate single-cell gel beads in emulsion (GEMs). scRNA-seq libraries were prepared with the Chromium Single Cell 3′ Library & Gel Bead Kit v.2 (10x Genomics, PN-120237). Final cDNA libraries were checked for quality and quantified using 2100 Bioanalyzer (High Sensitivity DNA Assay; Agilent Technologies). Libraries were sequenced using Illumina HiSeq 4000 system in Paired-End mode with 100 or 28 bases for read 1 and 100 bases for read 2, following Illumina's instructions. Image analysis and base calling were performed using RTA version 2.7.2 and Cell Ranger version 3.0.2.

## Single-cell RNA sequencing: analysis

### Primary analysis

FastQ files of each sample were processed with Cell Ranger count pipeline (10 X Genomics) version 6.1.1, that performs alignment, filtering, barcode counting, and UMI counting, using the pre-processed *Homo sapiens* reference GRCh38-2020-A (GENCODE v32/Ensembl 98) from 10 X Genomics. Data were then aggregated using Cell Ranger aggr tool which normalizes counts to the same sequencing depth and then recomputing the feature-barcode matrices and analysis (dimensionality reduction, *i.e.* UMAP, and clustering) on the combined data (24,651 cells).A subset of cells expressing endoderm or mesoderm markers (1662 cells, corresponding to 6.7% of the total population) and a subset of cells expressing high levels of stress markers (4399 cells, corresponding to 17.8% of the total population) were excluded, and analysis (dimensionality reduction and clustering) was performed again after exclusion of these non-neural and/or sub-optimally differentiated cell populations using Cell Ranger reanalyse tool, resulting a set of 18,590 cells regrouped in 15 clusters. More information about Cell ranger software can be found on the manufacturer website (https://support.10xgenomics.com/single-cell-gene-expression/software/overview/welcome).

### Trajectory analysis

Count data were normalized with the log normalize method using Seurat R package version 4.0.5 then filtered by keeping only genes with more than 5 UMI counts detected in at least 10 cells. Four analyses were performed: one with all cells (clusters 1–15), final data used in the analysis contained 18,590 cells and 7133 genes; one for ventral progenitors (only WNTi + FGF8 cells belonging to clusters 6, 7, 8, 9, and 13) with 4608 cells and 5382 genes; one for dorsal progenitors (only WNTi cells belonging to clusters 1, 2, 3, 4, 5, 12, 13, 14, and 15) with 8246 cells and 5827 genes; and the last one for all progenitors (dorsal + ventral; cells belonging to clusters 2, 5, 8, 9, 12, 13, 14, and 15) with 8876 cells and 6452 genes. Trajectory inference analyses were performed using slingshot method (*Street et al., 2018*) implemented in the dyno package (*Saelens et al., 2019*).

### VoxHunt analysis

Similarity maps between single-cell data and the public Allen Developing Mouse Brain Atlas expression data, stage E13, were computed using the VoxHunt R package version 1.0.1 (*Fleck et al., 2021*).

### Analyses of differentially expressed genes (DEGs) and Gene enrichment

Count data were normalized with the log normalize method using Seurat R package version 4.0.5 before performing the analysis (*Butler et al., 2018*). Differential gene expression analyses, realized with Seurat, between clusters of interest are performed using a Wilcoxon test whose performance for single-cell differential expression analysis has been evaluated in a previous report (*Soneson and Robinson, 2018*) and resulting p-values were adjusted for multiple testing using a Bonferroni correction. Enrichment analyses were performed on differentially expressed genes previously identified using cluster Profiler R package version 4.2.0 with Gene Set Enrichment Analysis (GSEA) method (*Subramanian et al., 2005*). Genes were ranked by their log2 Fold-Change. Enrichment analyses are

performed on the three domains of GO (Gene Ontology) terms: biological process; molecular function; and cellular component.

### Cluster annotation with SingleR

Annotation of Neurones_CTRL (clusters 1, 3, and 4 of WNTi cells), Neurones_FGF8 (clusters 1, 3, and 4 of WNTi + FGF8 cells), Progeniteurs_CTRL (clusters 2, 5, 12, 14, and 15 of WNTi cells) and Progeniteurs_FGF8 (clusters 2, 5, 12, 14, and 15 of WNTi + FGF8 cells) cells at the cluster and cell levels were performed with SingleR version 2.0.0 R package (*Aran et al., 2019*) using a previously published primary cell dataset (*Speir et al., 2021*). In this reference dataset, only cells corresponding to 16 PCW age and not to hippocampus area were kept. Before performing the annotation both datasets were normalized using the LogNormalize method implemented in the Seurat R package version 4.3.0 (*Hao et al., 2021*). *Figure 7—figure supplement 1C* shows a heatmap of the SingleR assignment scores as well as the corresponding inferred annotation for the clusters/cells. Scores allows users to inspect the confidence of the predicted labels across the dataset. Ideally, each cell/cluster (*i.e.* column of the heatmap) should have one score that is obviously larger than the rest, indicating that it is unambiguously assigned to a single label. A spread of similar scores for a given cell/cluster indicates that the assignment is uncertain, although this may be acceptable if the uncertainty is distributed across similar cell/cluster types that cannot be easily resolved.

## Acknowledgements

We thank Dr Micha Drukker for the kind gift of HMGU1 human iPS cells, and the GenomEast platform in Strasbourg, France, for the Illumina HiSeq 4000 Sequencing. We are grateful to Samah Rekima for the technical help with the iBV histology facility, and we also thank the iBV PRISM Microscopy facility for their regular support. This work was funded by the French Government (National Research Agency, ANR) through the 'Investments for the Future' programs IDEX UCAJedi ANR-15-IDEX-01, by the "Fondation pour la Recherche Médicale (Equipe FRM2020)" (#EQU202003010222), "Fondation de France" (#00123416), ERA-NET NEURON JTC21(ANR-21-NEU2-0003-03) grants to MS, by the "France Génomique" consortium (#ANR-10-INBS-0009) to MJ and by INSERM funding ("Dotation exceptionnelle prise de fonctions") to MB.

## Additional information

### Funding

| Funder | Grant reference number | Author |
| --- | --- | --- |
| Agence Nationale de la Recherche | IDEX UCAJedi ANR-15-IDEX-01 | Michèle Studer |
| Fondation pour la Recherche Médicale | EQU202003010222 | Michèle Studer |
| Fondation de France | 00123416 | Michèle Studer |
| ERA-NET NEURON JTC21 - Agence Nationale de la Recherche | ANR-21-NEU2-0003-03 | Michèle Studer |
| Agence Nationale de la Recherche | ANR-10-INBS-0009 | Matthieu Jung |
| Institut National de la Santé et de la Recherche Médicale | Dotation exceptionnelle prise de fonctions | Michele Bertacchi |

The funders had no role in study design, data collection and interpretation, or the decision to submit the work for publication.

## Author contributions
Michele Bertacchi, Conceptualization, Data curation, Formal analysis, Validation, Investigation, Visualization, Methodology, Writing – original draft, Project administration, Writing – review and editing; Gwendoline Maharaux, Data curation, Investigation, Methodology; Agnès Loubat, Data curation, Methodology; Matthieu Jung, Data curation, Formal analysis, Investigation; Michèle Studer, Conceptualization, Resources, Data curation, Supervision, Funding acquisition, Validation, Investigation, Writing – original draft, Project administration, Writing – review and editing

## Author ORCIDs
Michele Bertacchi ⓘ http://orcid.org/0000-0002-4402-4974
Gwendoline Maharaux ⓘ http://orcid.org/0009-0006-7076-5887
Matthieu Jung ⓘ http://orcid.org/0000-0002-3272-7322
Michèle Studer ⓘ https://orcid.org/0000-0001-7105-2957

## Decision letter and Author response
Decision letter https://doi.org/10.7554/eLife.98096.sa1
Author response https://doi.org/10.7554/eLife.98096.sa2

# Additional files

## Supplementary files
• MDAR checklist

## Data availability
The raw data from the single-cell RNA sequencing (scRNA-seq) experiments have been deposited in the NCBI Gene Expression Omnibus (GEO) and are publicly available under the accession number GSE276558. Further details can be accessed at the linked repository. Additional data for the graphs of immunostaining pixel intensity, cell counting, or real-time qRT-PCR are provided as Source data linked to the images.

The following dataset was generated:

| Author(s) | Year | Dataset title | Dataset URL | Database and Identifier |
| --- | --- | --- | --- | --- |
| Bertacchi M, Jung M, Studer M | 2024 | FGF8-mediated gene regulation affects regional identity in human cerebral organoids | https://www.ncbi.nlm.nih.gov/geo/query/acc.cgi?acc=GSE276558 | NCBI Gene Expression Omnibus, GSE276558 |

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
