## [Editor Report]

This study is of interest to neuroscientists interested in brain development, particularly human brain development. Human organoids are excellent models to investigate the relevance of gene and pathways in the context of embryonic development. In this important research paper, the authors present convincing evidence supporting a role of FGF8 in brain development.

---

## [Decision Letter]

[Editors' note: this paper was reviewed by Review Commons.]

---

## [Author Response]

Reviewer #1Evidence, reproducibility and clarity: Interesting results from exposing human brain organoids to FGF8 include suggestions that FGF8 contributes to the anterior to posterior patterning of the neocortex, as previously reported in mouse. Good, varied methods with reproducibility described well in the methods section. It would improve the reader's experience however to cite numbers of organoids used in specific experiments/assays in the main text.

We thank the Reviewer for the positive assessment of our study, and we agree that citing the number of organoids per experimental approach would better allow the readers to appreciate the intrinsic variability of organoid protocols. We have included the number of organoids per experiment in figure legends.

Full review:Organoids do not develop individual neocortical areas. To approach this issue of area identity, however, the authors compared control and FGF8-treated organoids against an existing dataset of transcriptomes of human fetal brains that separated pre-frontal, motor, somatosensory, and visual areas. This seems a good idea, but results showed both treated and untreated organoids alike expressed genes characteristic of somatosensory and pre-frontal cortical regions (anterior and midlevel areas) apparently suggesting that exogenous FGF8 had little effect. Because the previous dataset was not the authors' work, however, and because a comparison between organoids and actual human tissue is hard to interpret, this whole section is probably only confusing to include.

We would like to clarify to the reviewer that the effect of FGF8 on antero/posterior area identity is modest in our organoid system, suggesting that different doses or temporal windows of FGF8 treatment may be necessary to achieve a stronger modulation of area identity genes (as stated in Discussion). We agree with the Reviewer that, due to this moderate effect, the transcriptomic comparison with fetal brain areas might be confusing for readers. Therefore, we have moved these types of data to the Supplementary Material (new Figure 7—figure supplement 1). We thank the Reviewer for bringing this to our attention.

The authors further stress a dorsal/ventral effect in FGF8-treated organoids. The population of ventral telencephalic interneurons, produced in the lateral ganglionic eminence in mice, expand in the human organoids at the expense of glutamatergic neurons of the dorsal telencephalon. This may be consistent with the loss of ventral telencephalic structures in FGF8-deficient mice. The authors suggest that FGF8 expansion of interneurons is a novel finding not previously seen in animal research and may point to a human-specific characteristic. Readers may believe this part of the paper requires more support, just because multiple studies of FGF8 have not revealed this action. Overall, this paper would benefit from shortening, and by statements that some of the results suggest, but do not guarantee, particular conclusions.

We agree with the reviewer that before stating that FGF8-induced expansion of interneurons in dorsal telencephalic territories is a human-specific characteristic, more support in mouse studies would need to be performed. However, as stated by reviewer 2 below, there is some evidence that some ventral interneuron markers, such as ASCL1 and DLX2, are expressed in the dorsal telencephalon of the early fetal human cerebral cortex, even if at much lower levels than in ventral telencephalon, and that individual human cortical progenitors can generate both excitatory neurons and cortical interneurons in culture. Thus, FGF8 might promote an intrinsic capacity of dorsal cortical neurons to induce the production of ventral LGE-types of interneurons, which would indeed be a human (or maybe primate)-specific trait. We have better discussed this issue in the revised version of the manuscript, where we now state that “the FGF8-mediated induction of a ventral LGE identity in organoids could reflect a human-specific trait: the inherent ability of human glutamatergic progenitors to generate GABAergic neurons with an LGE-like molecular signature.”.

SignificanceThe paper is for a fairly specialized audience interested in the development of the cerebral cortex, but also has interest regarding developmental human brain defects

Although the manuscript sounds upon first reading specific to a specialized audience interested in cortical development, we believe that the strength of our human organoid system is the formation of regionalized organoids including brain regions other than the cortex. Moreover, considering the increasing attention on brain organoids in general, and the lack of information on the action of FGF8 during human cortical development, we are confident that this study will attract a broader audience.

Interesting results from exposing human brain organoids to FGF8 include suggestions that FGF8 contributes to the anterior to posterior patterning of the neocortex, as previously reported in mouse. Good, varied methods with reproducibility described well in the methods section. It would improve the reader's experience however to cite numbers of organoids used in specific experiments/assays in the main text.

We thank again the reviewer for acknowledging the potential of our study. As

previously mentioned, we agree that providing information about the number of organoids used has further supported the statistical analysis.

Reviewer #2Evidence, reproducibility and clarity……However, organoid technology offers a solution to this and the present study presents an elegant approach to addressing how FGF8 signalling directs both anterior/posterior and dorsal/ventral identity in neural progenitors and their offspring in human development. This has both biological and clinical relevance has the study demonstrates how FGF8 may be a key regulator of expression of susceptibility genes for neurodevelopmental conditions. The methods and approach are described clearly and in great detail and it serves as an exemplar for how studies like this might be pursued in the future. Likewise, the results are presented logically, using excellent figures with clear descriptions of the findings. It is positively entertaining to read and very thought provoking. We don't have any major issues with the conclusions.

We sincerely appreciate the reviewer’s enthusiastic and thoughtful feedback. The positive remarks on the clarity and detail of our methods and results are very encouraging, and we are pleased that the reviewer found our study both entertaining and thought-provoking.

We have some minor issues over presentation and interpretation that we would like the authors to consider.Developmental staging. It is stated that the organoids have reached a developmental stage equivalent to 16.5 GW based on expression of key genes such as CRYAB. Firstly, we would prefer an unambiguous way of stating age such as post-conceptional age. It is never clear what gestational weeks exactly means (post-menstrual, post-ovulatory?). Secondly, in several figures, UMAPs generated from the organoids are presented alongside representative mouse brain sections from E13.5 which is equivalent to about 11 post conceptional weeks in human. Although we find the mouse sections helpful, perhaps the potential discrepancy in developmental stage should be pointed out.

We agree with the reviewer that the staging of human organoids in vitro can be very tricky. We have clarified this issue by using post-conceptional weeks (PCW) instead of gestational weeks in the revised version of the manuscript. It is true, that schematic representations of brain sections of mouse telencephalon of around E13.5 were used in the paper, but the idea was to choose an age where dorsal and ventral territories are clearly separated during embryogenesis to highlight the expression of the different genes. Additionally, many developmental genes show clear gradients only early in development, while they tend to be uniformly expressed later on. For these reasons, we kept the current schematics as we believe they work nice to highlight the dorsoventral or anteroposterior character of key developmental genes next to scRNA-seq data. However, we now clearly state in figure legends that “brain schematics with gene expression patterns are based on embryonic day 13.5 stainings obtained from the Mouse Allen Brain Atlas.”, so that the readers will be aware of possible limitations when comparing gene expression between mouse embryos and human organoid cells.

Dorso-ventral patterning. Firstly, we wondered why VGLUT2 was used as a marker for dorsal identity when it is generally regarded as being expressed by subcortical neurons, e.g. thalamus and midbrain, whereas VGLUT1 is the standard marker for cortical neurons :https://doi.org/10.1016/j.tins.2003.11.005? Potentially, VGLUT2 expression may be more an indicator of mid/hindbrain identity than cortical identity. Is there any evidence for VGLUT2 expression by cortical cells in development? Also, MASH1 (more correctly called ASCL1) is not exclusively ventral, having shown to be expressed in a subset of intermediate progenitor cells for glutamatergic neurons in rodent BRITZ doi:10.1093/cercor/bhj168 and particularly human doi: 10.1111/joa.12971 ALZU’BI. We are surprised that the recent evidence that human cortical progenitors do have capacity to generate GABAergic neurons DELGADO 10.1038/s41586-02104230-7; 10.1101/2023.11.06.565899 NAN KIM is not mentioned in this section as perhaps FGF8 doesn't so much ventralise progenitor cells as promote an inherent property. This might explain why MGE-like identity is not observed, whereas LGE/CGE like is, as it has already been shown that MGE-like gene expression by dorsal progenitors is very much less likely than LGE/CGE like expression DELGADO 10.1038/s41586-021-04230-7; ALZU’BI 2017 DOI 10.1007/s00429-016-1343-5

We fully agree and thank the reviewer for bringing to our attention this interesting discussion and pointing to our confusion between VGLUT1 and VGLUT2 expression profiles. After checking our scRNA-seq data, we realized that the Reviewer is absolutely correct about the issue of using VGLUT2 as a dorsal telencephalic marker, as it is expressed in both dorsal and ventral cells. In contrast, VGLUT1 appears to be more specific for neocortical (dorsal) neurons (see Author response image 1). Moreover, it perfectly fits with our results showing a downregulation of VGLUT1 in dorsal glutamatergic neurons.

**Author response image 1. sa2fig1:** 

To support this point, we have validated the expression profile of VGLUT2 in dorsal cortical neurons, by performing triple VGLUT2/TRB1/CTIP2 and double VGLUT2/SATB2 stainings, that have been added in Supplementary material (see new Figure 4—figure supplement 2G-H’). This has allowed to confirm the use of VGLUT2 as a dorsal marker. We have also performed additional immunostainings involving VGLUT1, juxtaposed with GAD67 to assess dorso-ventral neuronal balance. This new analysis has been quantified using AI and integrated into Figure 4.Notably, these experiments have provided a comprehensive understanding of the expression patterns of VGLUT1 and VGLUT2 in the dorsal or ventral telencephalon and have further elucidated their utility as markers for specific neuronal populations in human brain organoids.

Furthermore, and importantly, we fully agree with the reviewer that human dorsal cortical progenitors do have the ability to generate GABAergic neurons, even if at lower efficiency than compared to glutamatergic neurons, and that FGF8 might promote this inherent property in human organoids. This new discussion and the new references suggested by the reviewer have significantly contributed to our data interpretation about LGE/MGE development. Again, thank you to the reviewer for these very insightful suggestions.

MEA recordings. The presentation of electrophysiological data is quite simple. Detection of spikes is claimed therefore representative traces of the spikes should be included and these can be easily generated with the Maxwell system software. It isn't clear how many times the experiments were repeated and there is no statistical analysis. For example, in the text they state on page 15 'Notably, WNTi+FGF8 organoids showed lower spike frequency (firing rate) and amplitude'. The amplitude difference is 43uV vs 41uV; we doubt this is significantly different. Threshold for detecting burst firing appears to be different between Figure 5C and 5d. Why? Shouldn't it be the same? The axonal tracking analysis in Figure 5E/F needs more explanation. How many axons were tracked? Is there any statistical analysis beyond means and standard deviation?

We agree with the Reviewer that the presentation of our electrophysiological data needed further improvement. We have repeated key recordings on 4 additional samples coming from 2 different batches, which have allowed us to conduct a comprehensive statistical analysis (see revised Figure 5).

In detail, we have:

Extracted representative traces of spikes from the Maxwell software, which have been included as Supplementary material (new Figure 5—figure supplement 1A). Footprints of action potentials have been extracted using the in-built analysis tool available in the software.

Performed axon tracking analysis on 3 control and 3 FGF8-treated samples coming from 2 distinct batches of organoids. Recordings and analyses have been conducted at different in vitro stages to monitor the growth of axonal tracts, enabling us to perform statistical analysis and observe the temporal evolution of axonal growth (see new Figure 5—figure supplement 1C-G).

Tested the effects of transient GABA-A receptor inactivation on WNTi+FGF8 organoids via Bicuculline treatment to demonstrate the specific functional effect of increased GABAergic activity upon FGF8 treatment (new Figure 5—figure supplement 2).

Furthermore, placing the threshold for detecting bursts in the network analysis at different levels in control or treated samples seems to be a routine procedure in this MEA system. Indeed, while the user can set a fixed multiplying factor (that is, of course, the same for both control and treated samples), it is the software that multiplies such factor by the basal average activity of the sample. In this way, bursts can be detected as synchronized activity emerging from the basal one, which, of course, varies in every sample. We have better explained this point in the Materials and methods section, and we thank the reviewer for raising this lack of clarity.

Anterior/posterior patterning. Returning to the subject of cortical GABAergic neurons, it has been proposed that the prefrontal cortex contains a relatively higher proportion of GABAergic neurons, although the mechanism for this has not been elucidated (see https://doi.org/10.1111/joa.13055 and references therein). Might higher anterior FGF8 specifying cortical progenitors to produce GABA neurons have a role in this?Authors: We thank the reviewer for citing this very interesting review. It is highly possible that FGF8 normally expressed anteriorly might have a role in inducing distinct GABAergic subtypes, such as Calretinin+ interneurons, which have been found to be more abundant in frontal cortices of the developing human fetal brain. Our organoids are too early in terms of the developmental stage to verify whether interneuron subtypes such as CalR+ are more or less represented. We have now added this very interesting point to our revised discussion, stating that the dual role played by FGF8 as an inducer of both ventral and anterior brain identity, opens “intriguing hypotheses about the role of FGF8 in bridging cortical areal identity with the local abundance and type of GABAergic interneurons that, instead of migrating from ventral regions, could be produced locally by FGF8-exposed cortical progenitors.”.Nomenclature. As this study principally presents data on mRNA expression levels it might be preferable to use italicised capitals for all gene names (except where referring to mouse genes). Also, common names are used in places and standard gene names in others, e.g. COUPTF1 is referred to NR2F1 but VGLUT1 is not referred to SLC17A7 (also see above re MASH1). It would be good to see everything standardised.

We appreciate the Reviewer for highlighting these discrepancies. We have now standardized gene names both in the text and figures accordingly.

SignificanceThis study involves a very imaginative use of organoids combined with a variety of approaches to test if fundamental principles of forebrain development, particularly cell specification and regional patterning, that we have learnt from mouse models are relevant to human brain development. It also has clinical relevance as it explores potential disruptions to development that leader to diseases of higher cognition, such as autism of schizophrenia. It is a very accessible manuscript that should have broad appeal. It makes several incremental additions to the field and points the way to future experiments in this area.

We sincerely thank the Reviewer's insightful comments and positive assessment of our study.

Reviewer #3Evidence, reproducibility and clarity:In the manuscript "FGF8-mediated gene regulation affects regional identity in human cerebral organoids" the authors used FGF8 to change cellular fate in human brain organoids. The experiments are well-performed and the authors used well-established protocols to generate brain organoids. The results clearly show that FGF8 addition induces an increase of diencephalon/midbrain markers (OTX2, EN2), suggesting that long-term FGF8 treatment can induce also posterior regional identities. These data are reinforced also by scRNAseq highlighting a possible mix of cellular identity.

We thank the reviewer for this encouraging report about our study highlighting the significance of our findings.

Main concern:The authors should start using FGF8 at later stages than day 19-21, in trying to maintain the forebrain identity.

As the reviewer correctly pointed out, the temporal window of FGF8 treatment seems of pivotal importance for the final outcome of regional identity acquisition. Indeed, early treatment with FGF8 at day 5 disrupts FOXG1 expression in organoids, as it was demonstrated in the previous version of Figure 1—figure supplement 1. In accordance with the reviewer’s request, we have treated organoids with FGF8 at later stages (starting at day 20) to test whether forebrain identity is maintained while midbrain induction is reduced when using this temporal window for FGF8 treatment.

To this purpose, we have treated organoids with the same amount of FGF8 but at different times to be able to compare the different treatments in parallel. In detail, we produced cortical organoids (using XAV-939 as a WNT inhibitor) and added 100 ng/ml FGF8 starting at day5 (early treatment), at day10 (normal treatment) or at day 20 (late treatment). Each condition has used at least n=6 organoids and all samples have been cultured until day 30. By immunostaining, we have measured the intensity of FOXG1 staining as a read-out of telencephalic identity and that of NR2F1 staining to evaluate FGF8 action. Interestingly, we found that adding FGF8 at day20 allowed to preserve high expression of FOXG1, supporting forebrain identity. However, this late FGF8 treatment also resulted in sub-optimal regulation of the major FGF8 target gene NR2F1, indicating that delayed FGF8 treatment could be less efficient in terms of FGF8 signalling modulation. We have incorporated these additional analyses into the Supplementary material (see new Figure 2—figure supplement 1) to provide a more comprehensive characterization of the efficiency time windows of FGF8. We also updated the main text, where we now state that “day20 FGF8 treatment was less efficient in modulating the FGF8-target gene NR2F1” and that day10-11 is “the most appropriate starting time point for FGF8 treatment, able to both preserve FOXG1 expression while efficiently modulating FGF8 target genes.”.

In conclusion, we thank the reviewer for raising this point. Not only because we think that this experimental setup has allowed us to further detail the effects of distinct temporal windows for FGF8 treatment (new Figure 2—figure supplement 1). But especially because these data have reconfirmed that the temporal window we selected for our study represents the most appropriate choice to effectively modulate FGF8 targets while maintaining high FOXG1 levels in human organoids.

To verify the identity of the neurons in the organoids the authors should check their ability to make projections in immunodeficient mice. Human iPSC-derived cortical neurons establish subcortical projections in the mouse brain after transplantation and the location of the different neuronal projections could reveal the rosto-caudal identity of the cortical neurons.

We agree with the reviewer that conducting in vivo transplants of human organoids would offer an interesting approach to testing the identity of differentiated neurons by tracking their projections. However, we believe that due to the multi-regional character of FGF8-treated organoids (which includes also midbrain-like neurons), their transplant into the neocortex would be of difficult interpretation and would not reveal the precise rostrocaudal identity of transplanted human cortical neurons, as requested by the reviewer. Furthermore, this would almost constitute an entire project on its own, given the technical challenges associated with such experimental approaches. We think that our thorough scRNA sequencing analysis is powerful enough for assessing cell identity, as supported by the majority of organoid studies investigating cell identity through scRNA-seq without resorting to transplantation. In our study, the scRNAseq analysis was subsequently validated by several steps of immunostainings, a simple but fundamental corroborative control approach that is sometimes overlooked in similar studies. Finally, we would like to emphasize that reviewers #1 and 2 found our complementary approaches (molecular, cellular, and functional) appropriate, well-performed, logical and reproducible.

Significance:The proposed protocol is useful to generate brain organoids with mixed cell populations from different regions of the brain (forebrain, midbrain, hindbrain). However, has limited applications since is not clear whether the proposed structures have some kind of organization.

We agree with the reviewer that each protocol comes with its own limitations and that a careful characterization of the proportion of different regional domains could definitively improve the significance and applicability of our protocol. To this aim, we are now using artificial intelligence-mediated detection of cortical versus midbrain-like domains in control and FGF8treated organoids, to further improve the characterization of distinct cellular populations and quantify the extent of their domains in multi-regional organoids. These data have been added in a new Figure (Figure 3—figure supplement 3).